# Gene isoforms as expression-based biomarkers predictive of drug response in vitro

Zhaleh Safikhani[1,2], Petr Smirnov [1], Kelsie L. Thu[1,3], Jennifer Silvester[1,3], Nehme El-Hachem[3], Rene Quevedo[1,2], Mathieu Lupien[1,2], Tak W. Mak[1,2,4], David Cescon[1,4,5] & Benjamin Haibe-Kains [1,2,6,7]

Next-generation sequencing technologies have recently been used in pharmacogenomic studies to characterize large panels of cancer cell lines at the genomic and transcriptomic levels. Among these technologies, RNA-sequencing enable profiling of alternatively spliced transcripts. Given the high frequency of mRNA splicing in cancers, linking this feature to drug response will open new avenues of research in biomarker discovery. To identify robust transcriptomic biomarkers for drug response across studies, we develop a meta-analytical framework combining the pharmacological data from two large-scale drug screening datasets. We use an independent pan-cancer pharmacogenomic dataset to test the robustness of our candidate biomarkers across multiple cancer types. We further analyze two independent breast cancer datasets and find that specific isoforms of IGF2BP2, NECTIN4, ITGB6, and KLHDC9 are significantly associated with AZD6244, lapatinib, erlotinib, and paclitaxel, respectively. Our results support isoform expressions as a rich resource for biomarkers predictive of drug response.

[1] Princess Margaret Cancer Centre, University Health Network, 101 College Street, Toronto, ON, Canada M5G1L7. [2] Department of Medical Biophysics, University of Toronto, 101 College Street, Toronto, ON, Canada M5G1L7. [3] Institut de Recherches Cliniques de Montréal, 110 Pine Avenue West, Montreal, QC, Canada H2W 1R7. [4] Campbell Family Institute for Breast Cancer Research, 620 University Avenue, Toronto, ON, Canada M5G2C1. [5] Division of Medical Oncology and Hematology, Department of Medicine, University of Toronto, 27 King's College Circle, Toronto, ON, Canada M5S 1A1. [6] Department of Computer Science, University of Toronto, 10 King's College Road, Toronto, ON, Canada M5S 3G4. [7] Ontario Institute of Cancer Research, 661 University Avenue, Suite 510, Toronto, ON, Canada M5G 0A3. Correspondence and requests for materials should be addressed to B.H-K. (email: bhaibeka@uhnresearch.ca)

Cell lines are the most widely used cancer models to study response of tumor cells to anticancer drugs. Not only have these cell lines recently been comprehensively profiled at the molecular level, but they have also been used in high-throughput drug screening studies, such as the Genomics of Drug Sensitivity in Cancer (GDSC)[1] and the Cancer Cell Line Encyclopedia[2]. The overarching goal of these seminal studies was to identify molecular features predictive of drug response (predictive biomarkers). Consequently, the GDSC and CCLE investigators were able to confirm a number of established gene–drug associations, including association between *ERBB2* amplification and sensitivity to lapatinib, and BCR/ABL fusion expression and nilotinib. They also found new associations such as *SLFN11* expression and response to topoisomerase inhibitors, thereby supporting the relevance of cell-based high-throughput drug screening for biomarker discovery. However, the biomarkers validated in preclinical settings are still largely dominated by genetic (mutation, copy number alteration, or translocation) as opposed to transcriptomic (gene expression) features. Therefore, there is a need for further investigation of transcriptomic markers associated with drug response in cancer.

The vast majority of pharmacogenomic studies investigated the association between gene-specific mRNA abundance and drug sensitivity[1–6]. However, it is well established that genes undergo alternative splicing in human tissues[7], and changes in splicing have been associated with all hallmarks of cancer[8]. Despite the major role of alternative splicing in cancer progression and metastasis[8], only a few small-scale studies have reported associations between these spliced transcripts (also referred to as isoforms) and drug response or resistance[9–14]. These limited, yet promising associations support the potential relevance of isoform

expression as a new class of biomarkers predictive of drug response. Among the mRNA expression profiling technologies, high-throughput RNA-sequencing (RNA-seq) enables quantification of both isoform and gene expression abundances at the genome-wide level. Recent studies have highlighted the advantages of RNA-seq over microarray-based gene expression assays[15–19]. In particular, microarray-profiling platforms are limited to pre-designed cDNA probes[15], and they depend on background levels of hybridization. They also suffer from limited dynamic range probe hybridization. As the detection of transcripts and genes using RNA-seq is based on high-resolution short-reads sequencing instead of probe design, they have the potential to overcome these limitations[17].

Recent initiatives have profiled hundreds of cancer cell lines using Illumina RNA-seq technology[3, 20–22]. As part of CCLE, the Broad Institute of Harvard and MIT released RNAseq profiles of 935 cancer cell lines through the Cancer Genomics Hub (CGHub[23], now moved to the NCI Genomic Data COmmon[24]), whereas Genentech deposited RNA-seq data for 675 cell lines on the European Genome-phenome Archive (EGA) as part of their Genentech Cell Line Screening Initiative (gCSI)[20, 22]. Two other initiatives used RNA-seq to profile panels of 70 (GRAY[3]) and 84 (UHN[21]) breast cancer cell lines. The availability of these valuable datasets offers unprecedented opportunities to further explore the transcriptomic features of cancer cells and study their association with drug response.

In this study, we explore the genome-wide transcriptomic landscape of large panels of cancer cell lines to identify isoform-level expression features predictive of drug response in vitro. On the basis of our new metaanalytical framework combining the GDSC and CCLE pharmacogenomic data for biomarker

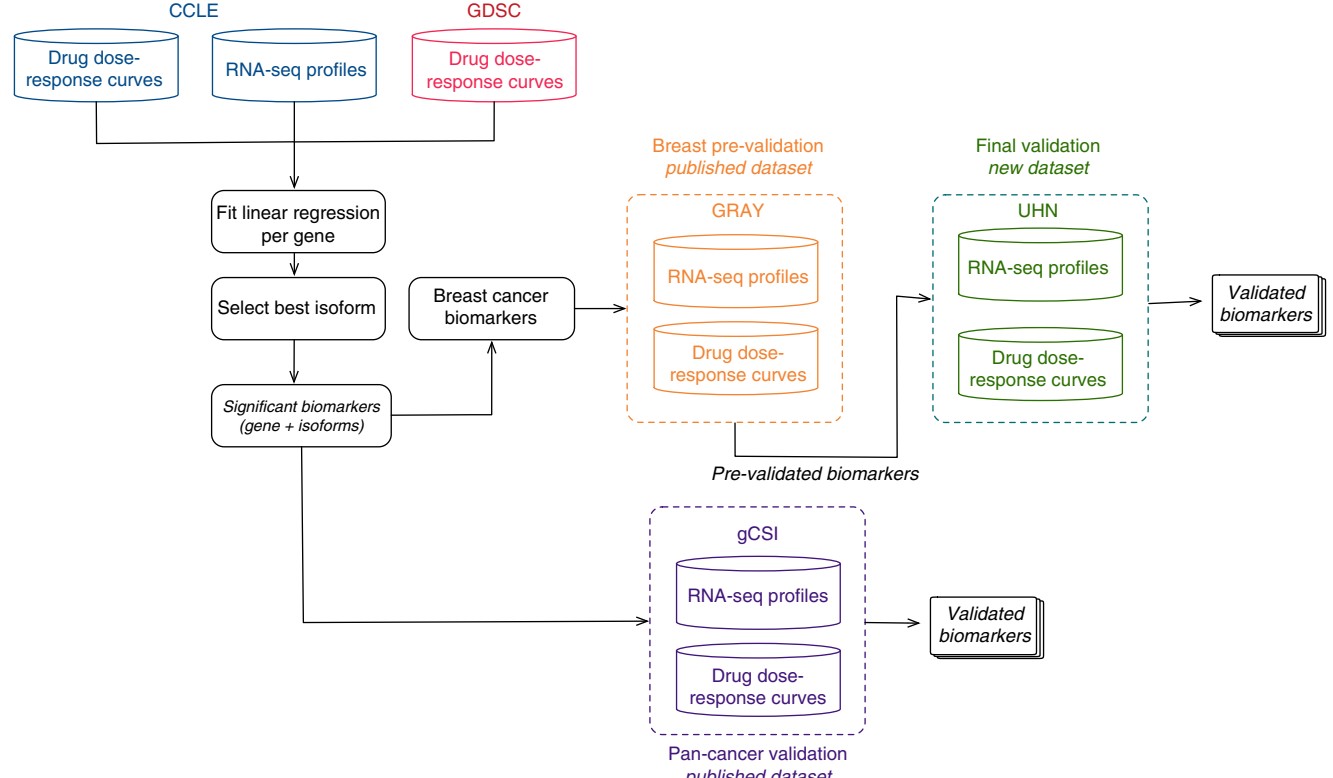

**Fig. 1** Analysis design of the study. CCLE (in blue) and GDSC (in red) are used to identify a set of biomarkers significantly associated with response to each of the 15 drugs screened in both training sets. Candidate biomarkers are tested both in pan-cancer (gCSI in purple) and in breast cancer-specific (GRAY in orange and UHN in green) settings. The validation in GRAY is referred to as pre-validation, enabling the selection of generalizable, isoform-based biomarkers for breast cancer. The newly generated UHN dataset is then used to test whether the selected isoform-based biomarkers are robust to the use of a different pharmacological assay (final validation)

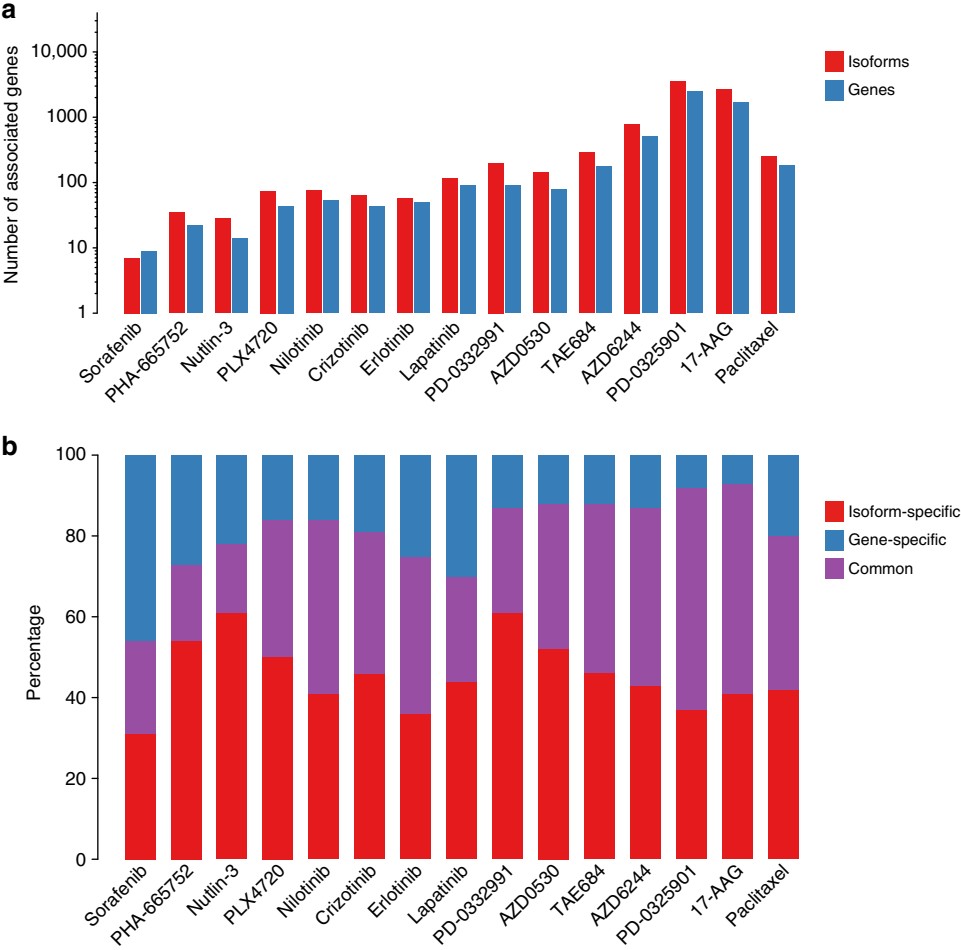

**Fig. 2** Comparison of number of statistically significant (FDR < 1%) predictive biomarkers for each of the 15 drugs in common between CCLE and GDSC. **a** Number of significant associations for gene and isoform models. The associations are compared at gene level, i.e., only one isoform is considered for each gene. **b** Proportion of biomarkers that are significant at the gene level, isoform levels, or both

discovery, we show that isoform-level expression measurements are more predictive of response to cytotoxic and targeted therapies than are gene-level expression values. We tested the accuracy of our most promising isoform biomarkers in an independent pan-cancer dataset, gCSI, and two breast cancer datasets, GRAY and UHN. We validated four isoform-based biomarkers predictive of response to lapatinib, erlotinib, AZD6244 (MEK inhibitor), and paclitaxel, indicating that isoforms constitute a promising new class of biomarkers for cytotoxic and targeted anticancer therapies.

## Results

**Discovery of isoform-based biomarkers.** We developed a meta-analysis pipeline enabling identification of molecular features predictive of sensitivity to 15 drugs (Fig. 1; Supplementary Table 1; Supplementary Fig. 1) across two large pharmacogenomics studies, namely CCLE and GDSC (Fig. 1). CCLE used the CellTiter-Glo (Promega) pharmacological assay, whereas GDSC used Syto60 (Invitrogen)[25], providing us with the opportunity to discover biomarkers generalizable to multiple measures of drug sensitivities. We identified a wide range of statistically significant biomarkers for each drug (10 to 1984 biomarkers with FDR <1% and concordance index > 0.55; Fig. 2a; Supplementary Data 1), with a significantly larger proportion of isoform-based biomarkers are predictive of drug response (Wilcoxon-signed rank test $p$-value $<10^{-3}$; Fig. 2a). For the majority of genes identified as

biomarkers, the highest ranking isoform, but not the overall gene expression, is significantly predictive of drug response (Fig. 2b). We investigated the discriminatory features between the "gene-specific", "isoform-specific", and "common" biomarkers by first assessing the number of alternatively spliced isoforms for each selected gene (Supplementary Fig. 2). As expected, the largest proportion of isoform-specific biomarkers originate from genes with multiple transcripts. We also categorized the predictive features by their biological types: protein coding, antisense, processed transcript, linc RNA, and pseudogenes (Supplementary Fig. 3). Although the largest proportion of protein-coding biomarkers are isoformic (isoform-specific and common), the processed transcript biomarkers is dominated by isoform-specific ones. The pseudogene type contains the lowest ratio of isoform-specific biomarkers.

We further tested whether the number of associations with drug sensitivity was significantly larger for isoforms than gene, copy number alterations, or mutations. Concurring with the Drug Sensitivity Prediction DREAM challenge[6], we found that expression-based features were more significantly associated with sensitivity for most of the drugs (Fig. 3a). Overall, there were significantly more isoform-based biomarkers than gene expression and copy number alterations (one-sided Wilcoxon rank sum test $p$-value < 0.001; Fig. 3a) and their concordance index was superior (one-sided Wilcoxon rank sum test $p$-value < 0.001; Fig. 3b). However, even though there were more isoform-based

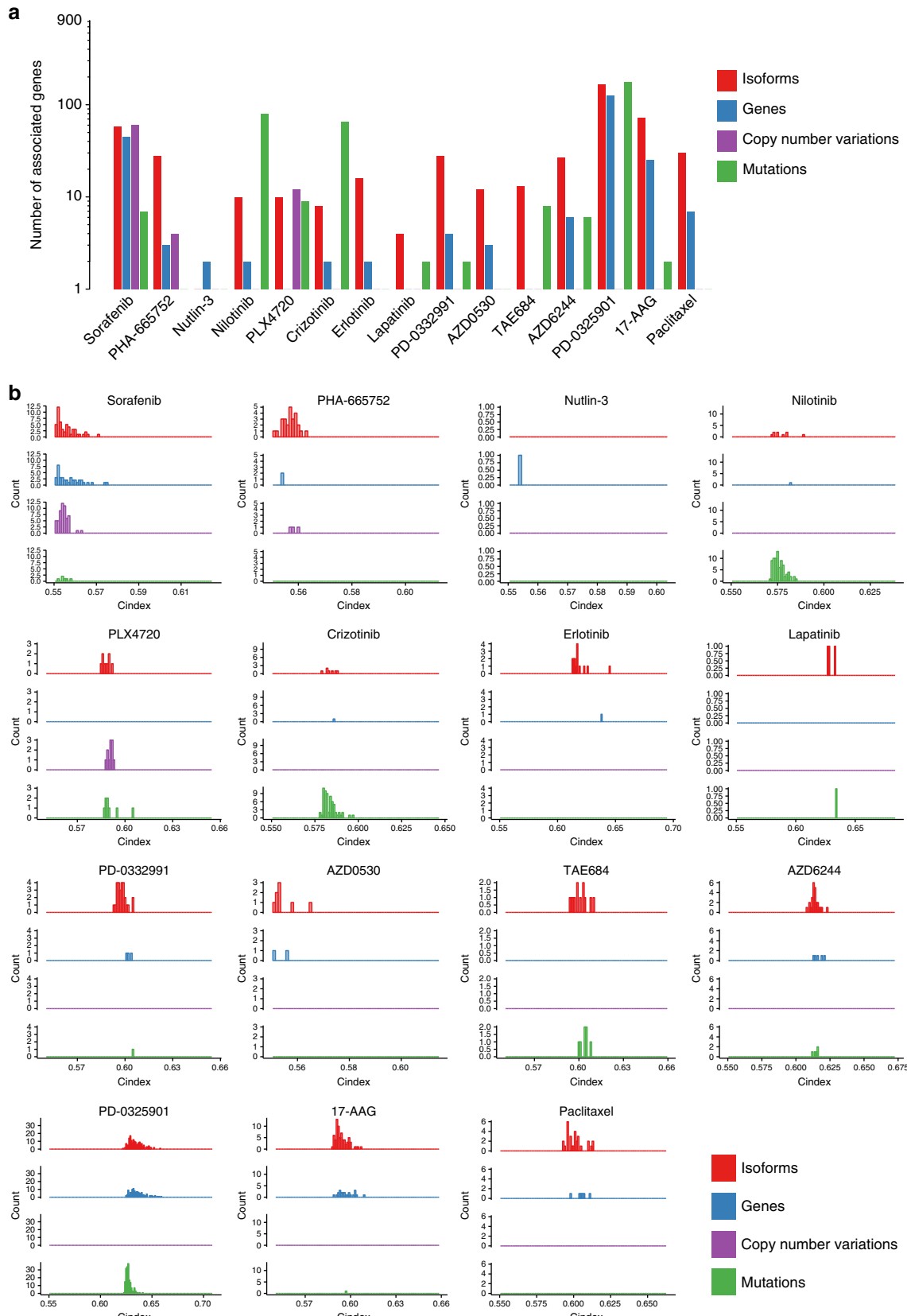

**Fig. 3** Comparison of significance and predictive value for multiple data types. The comparison is limited to 1600 genes for which mutations, copy number variation, and expression data were available. **a** Number of significant (FDR < 5%) biomarkers at the level of gene expression, isoform expression, mutation, and amplification. **b** Distribution of concordance indices (predictive value) of the significant biomarkers with respect to data type

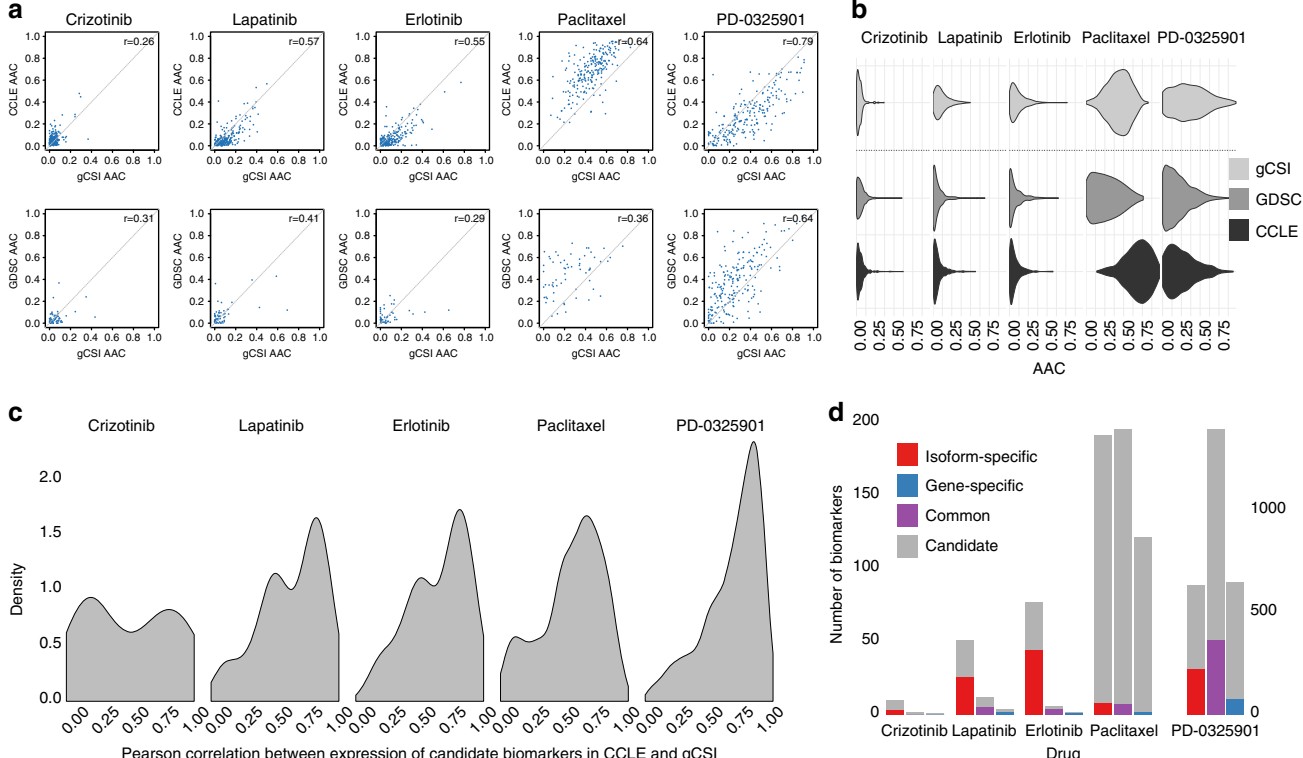

**Fig. 4** Pan-cancer validation of candidate biomarkers. Consistency between the training sets (CCLE and GDSC) and the validation set (gCSI) at the level of **a**, **b** drug sensitivity and **c** isoform and gene expression values. **a** The area above the drug response curve (AAC) values between the two training datasets and gCSI are plotted with their correlations for the common drugs. **b** The distributions of the AAC values compared for each and dataset. **c** The distribution of Pearson correlations for candidate biomarker expressions between CCLE and gCSI plotted for each common drug. **d** The number of candidate and pre-validated biomarkers plotted for each drug

biomarkers than mutation-based biomarkers for most drugs, mutations were more predictive for nilotinib, crizotinib, PLX4720 and TAE684 (Fig. 3). Although these results are limited to a set of 1600 genes for which all data types were available, they support isoform expression as a promising new class of biomarkers for the majority of the drugs in our training set.

**Pan-cancer validation of isoform-based biomarkers**. We first validated our candidate biomarkers identified in our training set using gCSI, a large pharmacogenomic dataset recently released by Genentech[20, 22]. This dataset contains 16 drugs, of which 5 are overlapping with our training set (Supplementary Fig. 4). Thanks to the large sample size of gCSI, we observed high validation rate for multiple drugs (61 and 54% for erlotinib and lapatinib, respectively; Fig. 4; Supplementary Data 2). Predicted and actual AAC values for the top biomarkers were reported for each tissue type in Supplementary Figs. 5, 6 and 16. These new results clearly indicate that reasonable validation rate can be achieved when a large validation set is available. We further investigated the cases of paclitaxel and the MEK inhibitor PD-0325901. Both drugs have broad growth inhibitory effects, as indicated by the large variance of drug sensitivity values (Fig. 4a, b). Concurring with our previous results[26], we observed that AAC values were inconsistent between the training sets (GDSC and CCLE; Supplementary Fig. 7), finding that extends to the comparison of GDSC and gCSI dataset ($\rho = 0.36$; Fig. 4a). On the contrary, PD-0325901 yielded moderate to high consistency in drug sensitivity data between the training sets ($\rho = 0.56$; Supplementary Fig. 7) and with validation set ($\rho = 0.64$ and 0.79 for GDSC vs. gCSI and CCLE vs. gCSI, respectively; Fig. 4a). In addition, the selected biomarkers tended to yield higher correlation between the

training and the test set for PD-0325901. These observations explains in part why the validation rate was low for paclitaxel, whereas the number of biomarkers was high for PD-0325901 (Supplementary Table 2).

**Pre-validation in an independent breast cancer dataset**. In vitro validation of drug response biomarkers in fully independent datasets has been shown to be challenging[27–30]. We therefore sought to assess the predictive value of our most promising isoform biomarkers for eight drugs screened both in our training sets and in the independent breast cancer dataset published by Daemen et al.[3] (referred to as GRAY), which used the same pharmacological assay as CCLE. We first selected the significant isoform-based biomarkers in our training set that were predictive in breast cancer cell lines (see "Methods"). We assessed the predictive value of these biomarker candidates in GRAY and tested whether these isoform biomarkers were significantly more predictive than their corresponding gene expression (Fig. 5; Supplementary Fig. 8; Supplementary Data 3). The validation success rate for isoformic biomarkers ranged from 0% (no validated biomarkers for crizotinib) to 25% validated biomarkers for paclitaxel (Supplementary Table 2). We found that the poor validation rate for crizotinib and sorafenib stems from inconsistency in their pharmacological profiles (Supplementary Fig. 9). On the basis of the number and effect size of biomarker candidates that were significant in GRAY, we selected AZD6244, lapatinib, erlotinib, and paclitaxel for further validation.

**Validation using a different pharmacological assay**. To test the robustness of our pre-validated biomarkers we generated a new set of drug sensitivity data combined with the RNA-seq profiles of

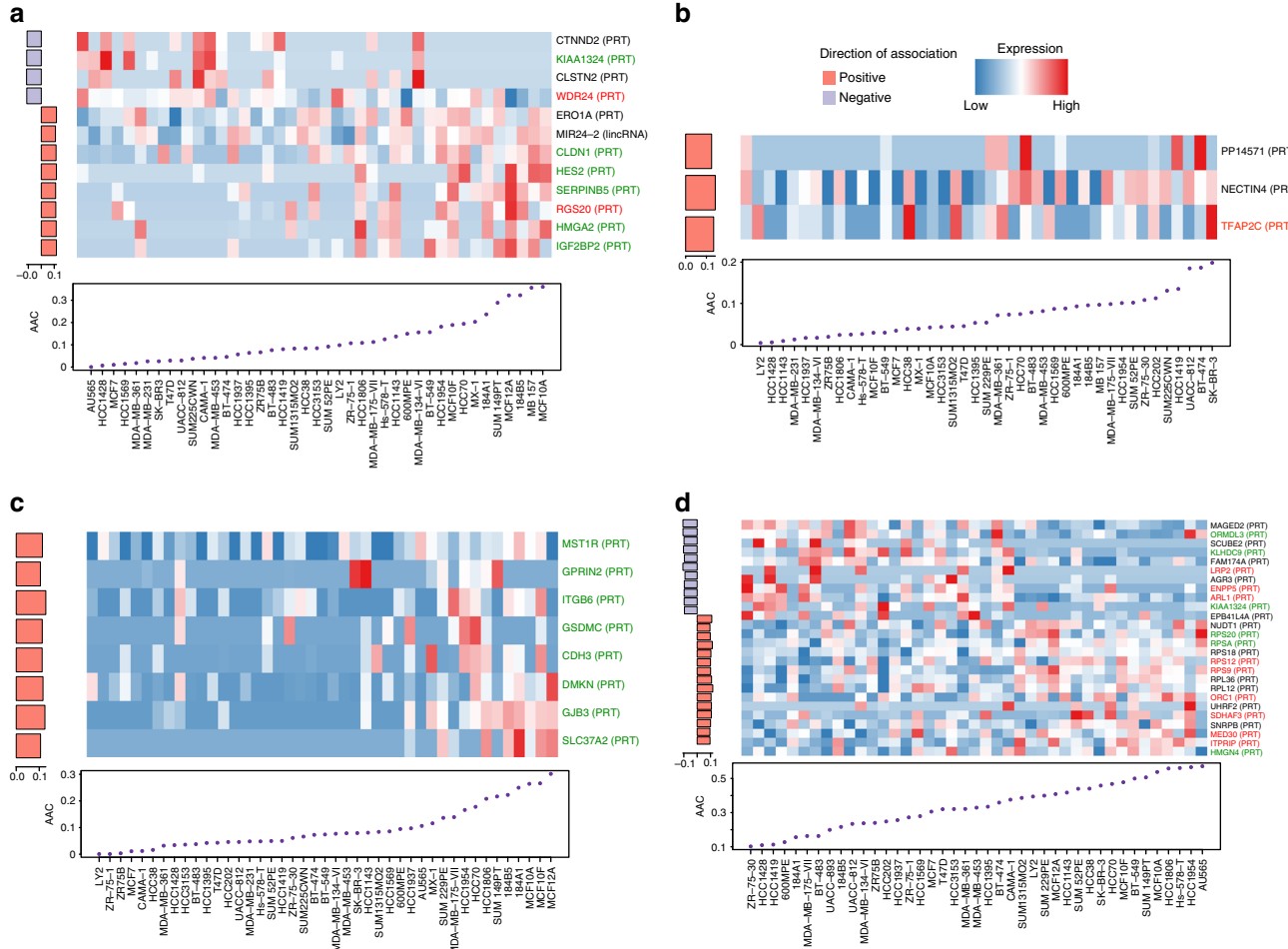

**Fig. 5** Isoform-based biomarkers successfully pre-validated in the independent GRAY dataset for **a** AZD6244, **b** lapatinib **c** erlotinib, and **d** paclitaxel. Cell lines are ordered by their sensitivity to the drug of interest and their isoform expression is shown in the heatmap, with the drug sensitivity (AAC) plotted below. The left side barplot shows the predictive value of the associations between isoform expression and drug sensitivity as the concordance index minus 0.50 multiplied by the sign of the coefficient in the corresponding regression model. Genes for which the candidate isoform is significantly more predictive than its corresponding overall gene expression values are represented in green, red, when overall gene expression is more predictive than isoform, and black otherwise

breast cancer cell lines published by Marcotte et al.[21] This new pharmacogenomic dataset is referred to as UHN. We screened cell lines with a different pharmacological assay (sulforhodamine B assay; SRB) from those used in the training and pre-validation sets. We first cultured cell lines to check their doubling time in a course of 120 h (Supplementary Table 3). Only cell lines with a growth rate/doubling time that was amenable to the 5-day SRB assay as a readout for cytotoxicity were considered for testing in the full 9-dose assay. We then assessed the anti-proliferative effect of cell lines to drugs using SRB assay in 96 well plates in triplicates. All the drug dose–response curves passed our quality controls (see "Methods").

Similar to the pre-validation performed in GRAY, we considered an isoformic biomarker to be validated if the linear association between its expression and drug sensitivity is both significant and in the same direction (same coefficient sign in the regression model). This resulted in validation of 2 out of 12, 2 out of 5, 6 out of 11, and 11 out of 35 biomarkers for AZD6244, lapatinib, erlotinib, and paclitaxel, respectively (Supplementary Table 2; Supplementary Data 4). We selected the isoform with the largest effect size for each drug (Supplementary Table 4; known biomarkers are provided as reference in Supplementary Table 5)) and investigated its exon occupancy and correlation compared

with the other isoforms of the same gene (Fig. 6; Supplementary Fig. 10). The selected *IGF2BP2-002* (ENST00000346192), *NECTIN4* (ENST00000368012), and *ITGB6-001* (ENST00000283249) isoforms were associated with sensitivity to AZD6244, lapatinib, and erlotinib, respectively (Fig. 6a–c; Supplementary Fig. 10), whereas the *KLHDC9-207* (ENST00000490724) isoform is associated with lack of sensitivity to paclitaxel (Fig. 6d; Supplementary Fig. 10). For *KLHDC9-207*, the predictive isoform was highly correlated with other isoforms of the same gene, sharing similar exon occupancy (Fig. 6h), whereas predictive isoform for *IGF2BP2-002*, *NECTIN-2014*, and *ITGB6-001* present a more specific expression pattern (Fig. 6e–g). Predicted and actual AAC values for these top validated biomarkers are provided in (Supplementary Fig. 11).

## Discussion
Although gene expression represents an important class of biomarkers for prediction of drug response in vitro[1–6, 22], association between gene isoforms and drug sensitivity has not been well studied despite the critical role of alternative splicing in cancer[8]. Our study is the first to describe a genome-wide meta-analysis of isoform-based biomarker predictive of drug response in vitro

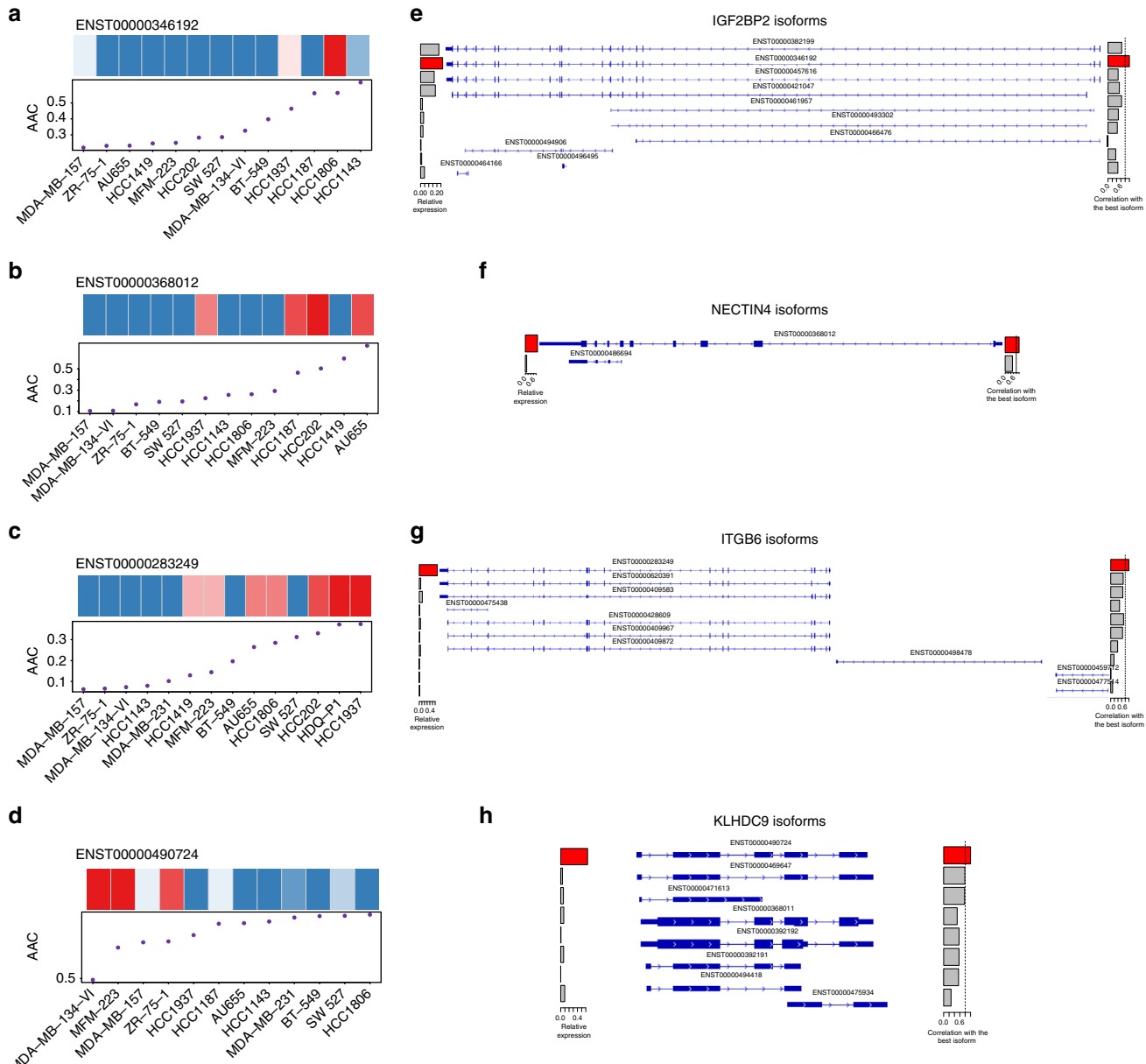

**Fig. 6** Validation of the candidate isoforms predictive of response to **a**, **e** AZD6244; **b**, **f** lapatinib; **c**, **g** erlotinib, and **d**, **h** paclitaxel in the independent UHN dataset generated where a different pharmacological assay (sulforhodamine B assay) was used to measure drug sensitivity. **a–d** Cell lines are ordered by their sensitivity to the drug of interest and their isoform expression is shown in the heatmap, with the drug sensitivity (AAC) plotted below. **e–h** Exon occupancy of each candidate isoform (*) is visualized using the Integrative Genomic Viewer, with a barplot on the right side representing the correlation ($\rho$) of expression between each isoform and the candidate isoform (red bar). A vertical dashed line represents $\rho = 0.8$ to identify highly correlated isoforms of the same gene

(Fig. 1). Controlling for the large number of isoforms, we found that significantly more genes had one of their isoforms predictive of response compared with overall gene expression for the vast majority of the drugs (Fig. 2a). Importantly, only a minority of biomarkers were solely predictive based on their overall gene expression and would have been missed by focusing on isoform expressions (Fig. 2b), supporting isoforms as a promising, untapped resource for drug response biomarkers.

Recognizing the challenges involved in biomarker discovery and validation from in vitro drug screening data[22, 25, 27, 28, 30–34], we further assessed the predictive value of our newly discovered isoform-based biomarkers for five drugs in the pan-cancer gCSI dataset, as well as four drugs (AZD6244, lapatinib, erlotinib, and

paclitaxel) in GRAY and UHN, two independent breast cancer pharmacogenomic datasets (Fig. 1; Supplementary Table 1). Thanks to the large sample size of gCSI, we obtained validation rates of over 50% for some targeted therapies (Supplementary Table 2). As expected, we found that consistency of pharmacological profiles between training set as well as the validation set was determinant to ensure reasonable validation rate. When testing the robustness of our biomarker candidate in breast cancer, we obtained low validation rates for eight drugs (33–51%; Supplementary Table 2) in our pre-validation phase, despite the fact that this study used the same pharmacological assay as CCLE to generate their drug sensitivity data (CellTiter-Glo; Supplementary Table 1). We found that many of the strongest

biomarkers were significantly more predictive of drug sensitivity at the isoform level compared to the overall gene expression level (Wilcoxon-signed rank test $p < 0.05$; Fig. 3).

Given that we and others have shown that the choice of pharmacological assay may influence drug sensitivity measurements[22, 25, 31], we sought to validate our candidate iso-form biomarkers using the sulforhodamine B assay (SRB), which differs from the assays used in the training and pre-validation datasets (Fig. 1). We selected 14 breast cancer cell lines and screened them with the set of four drugs. Despite the small sample size, we found 21 isoform biomarkers whose association is close to significance ($p < 0.1$; Supplementary Table 2). We selected the most predictive isoform for each drug to investigate its correlation with the other isoforms of the same gene (Fig. 6). AZD6244 is a highly selective *MAPK-ERK* kinase inhibitor. As a biomarker predictive of response to the *MEK* inhibitor AZD6244 in breast cancer, we identified ENST00000346192, one of the longest isoforms of the insulin-like growth factor 2 mRNA-binding protein 2 (*IGF2BP2*) (Fig. 6e), which codes a protein with 556 amino acids. It has been shown that *IGF2BP2* stabilizes and regulates many oncogenic proteins such as *NRAS*, *MAPK1*, and *RAF1* involved in the *NRAS-MEK-ERK-MAPK* signaling path-way, hyperactivated in many cancers[35]. As *IGF2BP2* lies upstream of the mitogen-activated protein kinase (*MAPK*)/extracellular signal-regulated kinase (*ERK*) pathway, this suggests that *IGF2BP2* could be a putative biomarker of sensitivity for com-pounds acting upstream of the *MAPK-ERK* pathway and target-ing *MEK1/2* such as AZD6244.

We investigated the association between isoform expressions and sensitivity to lapatinib, a dual tyrosine kinase inhibitor, which interrupts the *HER2/neu* and epidermal growth factor receptor (*EGFR*) pathways, and is a FDA-approved drug for breast cancer. In our study, *NECTIN4* isoform expression (ENSG00000143217) has been associated with sensitivity to lapatinib (Fig. 6b). It is the longest and the only protein-coding isoform of *NECTIN4* (Fig. 6f). It has been reported that *Nectin-4* is overexpressed in several human cancers, including lung, and breast cancer[36]. Furthermore, it has been shown that expression of *Nectin-4* strongly correlates with the basal-like marker *EGFR* in breast cancer, thus proposed as a putative new histological and ser-ological tumor-associated biomarker[37, 38]. This strengthens our hypothesis that the expression of *Nectin-4* isoform would sensi-tize a subset of breast cancer cells to lapatinib.

Our results also indicate that sensitivity to the *EGFR* inhibitor, erlotinib, can be predicted by the expression of the ENST00000283249 isoform of *ITGB6* (Fig. 6c). Previous reports showed that the interaction between integrins and several receptor tyrosine kinases like *HER2* and *EGFR* leads to a crosstalk between signaling events that have been implicated in tumor progression and metastasis[39, 40]. Desai et al.[41] recently identified a significant positive correlation between *ITGB6* and *EGFR* pro-teins, suggesting that *ITGB6* and *EGFR* are interacting, and that *ITGB6* isoform expression sensitizes cells towards erlotinib, an *EGFR* inhibitor.

Lack of sensitivity or innate resistance to chemotherapies is a major issue in current breast cancer management[42]. We found that expression of the ENST00000490724 isoform of the kelch domain-containing 9 (*KLHDC9*) gene is associated with lack of sensitivity to paclitaxel in breast cancer cell lines (Fig. 6d). Paclitaxel is a known microtubule stabilizer that inhibit cell division (mitosis) and is a broad chemotherapeutic agent. *KLHDC9* has been characterized as an interaction partner of cyclin *A1/CDK2* complex, a key regulator of the mitotic cell cycle[43]. Furthermore, *KLHDC8B*, another member of the Kelch domain-containing protein family has been implicated in mitotic regulation and chromosomal segregation in lymphoma[44]. We

hypothesize that a lower expression of *KLHDC9* isoform will negatively affect mitotic integrity, potentiating the activity of paclitaxel on rapidly dividing cells. However, additional char-acterization of the biology underlying the isoform specificity of this association are required to draw firm conclusion on the functional relation between *KLDHC9* and pacliatxel resistance (Fig. 6h).

This study has several potential limitations. First, our bio-marker discovery pipeline is restricted to univariate linear asso-ciation between gene and isoform expression and drug sensitivity. These two restrictions have been imposed to mitigate the risk of overfitting as the development of multivariate, potentially non-linear predictors of in vitro drug sensitivity has been proven to be challenging[27, 28]. A larger compendium of pharmacogenomic datasets will be necessary to overcome this. A second limitation lies in the use of a single processing pipeline to quantify expression of each individual transcripts from Illumina RNA-seq data. We choose to use the HISAT2/StringTie pipeline for RNA-seq[18] because of its excellent performance in recent benchmarks[45, 46]. We recognize that many alternatives exist[47–49] but their comparison is out of the scope of the present study. Third, we used a bootstrapping procedure, coupled with a Wilcoxon-signed rank test to statistically compare our candidate biomarkers to a model including tissue type as sole predictor. We recognize that the dependency across bootstraps may inflate the $p$-values. New statistical testing procedures are yet to be devel-oped to properly address this issue. Fourth, the tissue-specific validation of our biomarkers is limited to breast cancer cell lines, the only tissue type for which we had multiple independent pharmacological and molecular datasets available. Lastly, we are aware that clinical relevance of our candidate isoform biomarkers remains to be tested in vivo, using patient-derived xenografts and clinical samples in both retrospective and prospective studies. The release of RNA-seq and pharmacological profiles of such samples will allow to further challenge our findings in future studies.

The advent of RNA-sequencing technology enables efficient quantification of alternatively spliced transcripts in cancer cells. Our genome-wide search for biomarkers demonstrates that gene isoforms constitute a rich resource of transcriptomic features associated with response to targeted and chemotherapies in vitro. We found that specific isoforms of IGF2BP2, NECTIN4, ITGB6, and KLHDC9 were significantly associated with AZD6244, lapatinib, erlotinib, and paclitaxel, respectively, in multiple screening using different pharmacological assays. Our results suggest that isoform-based biomarkers are more frequent and more significantly associated with drug sensitivity than overall gene expression, opening new avenues for future biomarker dis-covery for in vitro and in vivo drug screening.

## Methods

**Published pharmacogenomics studies**. We used our PharmacoGx platform[50] to create curated, annotated, and standardized pharmacogenomic datasets, which comprises CCLE[2], gCSI[22], GDSC[1], and GRAY[3] (Supplementary Table 1). CCLE, gCSI, and GRAY pharmacological data were generated using the CellTiter-Glo assay (which quantitates ATP, Promega), whereas GDSC used the Syto60 assay (a nucleic acid stain, Invitrogen)[25]. We updated CCLE, gCSI, and GRAY PharmacoSets to include gene and isoform-level expression data processed from the raw RNA-seq profiles downloaded from CGHub[23] and NCBI GEO[51], respectively.

**RNA-seq data processing**. The RNA-seq reads were aligned to the Ensembl Genome Reference Consortium release GRCh38[52] using HISAT2[53] and StringTie[45, 46] was used to annotate genes and isoforms, and quantify their expression. Gencode version 25[54] was used as the transcript model reference for the alignment as well as for all gene and isoform quantifications. Gencode annotated a total of 58,037 genes, which includes 19,950 protein-coding genes, 15,767 long noncoding RNA's (lncRNA's), and 14,650 pseudogenes. Expression values were computed as the $\log_2(\text{FPKM}+1)$[45], where FPKM represents the number of

fragments per kilobase per million mapped reads units that control sequence length and sequencing depth[55].

**Pharmacological data processing**. We developed a unified framework to process the raw pharmacological data of CCLE, GDSC, gCSI, and GRAY and to obtain the drug dose–response curves using a standard curve fitting algorithm[50] (Supplementary Methods). To summarize the drug dose–response curves into a single-sensitivity measure, we computed the area under the curve (AUC) metric, which combines both potency and efficacy of drug responses[56] (Supplementary Fig. 12; Supplementary Methods). Compared with $IC_{50}$ and $E_{max}$ metrics, which represent only one point on the drug dose–response curve, AUC values are computed by integrating all data points. Consequently, AUC has been shown to be more reproducible across pharmacogenomic studies[28, 31]. In this study, we used the area above the drug dose–response curve (AAC = 1−AUC; Supplementary Fig. 12) so that higher AAC values represent higher drug sensitivity.

**Biomarker discovery**. To identify gene and isoform expression robustly associated with drug sensitivity, we developed a machine learning pipeline combining linear regression models with a bootstrapping procedure for stringent model selection. Our choice of model assumes a linear relationship between molecular features and drug responses. Although violation of this assumption may result in biased predictions, linear models are robust to variation or noise in the data, making them less prone to overfitting in a high-dimensional context such as pharmacogenomics. Therefore, the association between each molecular feature and response to a given drug is assessed by fitting linear models using the gene or isoform expression across cell lines as predictor variables, adjusted for tissue of origin of cancer cell lines, and their sensitivity values to the given drug as dependent variables (Supplementary Fig. 13). To assess the association of each gene and its isoforms to a given drug, three linear models were constructed for each dataset as following:

$$M_0 : Y = \beta_0 + \beta_T T, \qquad (1)$$

$$M_1 : Y = \beta_0 + \beta_T T + \beta_G X_G, \qquad (2)$$

$$M_2 : Y = \beta_0 + \beta_T T + \beta_I I_G \quad \forall I_G \in G_I, \qquad (3)$$

where $T$ represents the tissues of origin as a vector of size $N \times 1$; $N$ is the number of cell lines; $Y$ denotes the drug sensitivity vector of size $N \times 1$ containing the drug sensitivity values (AAC) of the cell lines treated by the drug of interest; $X_G$ represents a vector of size $N \times 1$ of $\log_2$ normalized FPKM values for the expression of gene G across all the cell lines; $G_I$ is all the isoforms of gene G; $I_G$ is a vector of size $N \times 1$ of $\log_2$ normalized FPKM values for each isoform of G across all the cell lines. The effect size of each association is quantified by $\beta_G$ and $\beta_I$, which indicate the strength of associations between drug response and the molecular feature of interest, adjusted for tissue type. To estimate standardized coefficients from the linear model, the variables $Y$ and $X_G$ and $I_G$ are scaled (standard deviation equals to one, mean equals to zero). The null model (Eq. (1)) estimates the association between drug response and tissue source, as we previously showed that drug sensitivity in vitro is tissue specific[57]. The models in Eqs. (2) and (3) estimate the strength and significance of the association between drug sensitivity and the gene level and its best isoform expressions, respectively.

To address the lack of reproducibility of drug sensitivity measurements across studies[26, 31], we developed a meta-analytical pipeline to combine the pharmacological data from CCLE and GDSC. The June 2014 release of CCLE consists of 11,670 experiments in which 24 drugs have been screened on 1053 cancer cell lines from 24 tissue origins. GDSC release 5 comprises 79,903 experiments for 140 different drugs tested on a panel of up to 778 unique cell lines from 30 tissue types. The panel of drugs and cell lines screened in these two datasets overlapped for 15 compounds and 706 cell lines, respectively (Supplementary Data 5 and 6; Supplementary Fig. 1). Univariate gene–drug associations were computed using the linear models described in above-mentioned equations with CCLE RNA-seq data as predictors and CCLE and GDSC drug sensitivity data separately. We recognize that using CCLE RNA-seq data in combination with GDSC is suboptimal as gene expression of cell lines are subject to biological and technical variations[33]. In the absence of RNA-seq data for GDSC, we could only address the variations observed in the drug sensitivity measurements, which we demonstrated to be significantly higher than variations in gene expression data[26]. To ensure that cell line identity was conserved across CCLE and GDSC, we performed SNP fingerprinting (Supplementary Methods) and filtered out the cell lines identified as different across studies using a cutoff of 80% concordance[26]. In addition, we compared the microarray expression profiles of cell lines between microarray and RNA-seq profiles, which resulted in good concordance (Supplementary Fig. 14) supporting that expression profiling are consistent.

To determine the most predictive isoform for each gene the predictive value (concordance index[58]) of all of its isoforms is estimated with Eq. (3) and the most significant isoform (the one with the smallest Bonferroni-corrected p-value) is selected for further analysis (Supplementary Fig. 13). Comparison of the predictive value of each model was performed using a bootstrapping procedure: 100

resampled datasets are generated, where the cell lines are obtained by sampling with replacements from all the cell lines with sensitivity and expression profile available for a given drug. The linear regressions are solved for each bootstrap using the resampled set (~2/3) and unselected cell line set (~1/3) for training and testing, respectively. To evaluate the prediction performance of a gene or isoform model, its vector of concordance index values is compared with a null model using a one-sided Wilcoxon-signed rank test. Bootstrapping procedure is applied on the gene and its most predictive isoform. We recognize that the dependence between the different test sets may inflate the Wilcoxon's test p-values; however such bias would be present for both genes and isoforms, resulting in a fair comparison. To combine the fitted models obtained from CCLE and GDSC, their coefficients and p-values were averaged and weighted by the number of cell lines in those datasets (Supplementary Fig. 13). To control for multiple testing, we corrected the p-values obtained for all genes and isoforms, separately, using the false discovery rate (FDR) method[59].

**Pan-cancer validation of isoform-based biomarkers**. Beside CCLE, the recent gCSI pharmacogenomic study[20, 22] is the only publicly available dataset including RNA-seq and pharmacogenomic profiles for a large panel of cell lines spanning across multiple cancer types (Supplementary Fig. 4). We therefore used gCSI to validate our candidate biomarkers in a pan-cancer setting. We computed the significance of the linear association between the biomarker expression and drug response controlled for tissue type (unadjusted p-value < 0.05) with the same direction of association (sign of the coefficient $\beta$) as the training sets. Similarly to the discovery phase, we selected the validated biomarkers by statistically comparing the concordance index distribution of the isoform-based and gene-based models to concordance indices of the null model (tissue type only) based on the bootstrap procedure using a one-sided Wilcoxon-signed rank test (Supplementary Fig. 13).

**Pre-validation of isoform-based biomarkers in breast cancer**. We also validated the accuracy of our biomarkers using a previously-published independent dataset, GRAY[3], which includes RNA-seq of a panel of 70 breast cancer cell lines screened with 90 FDA-approved drugs (CellTiter-Glo pharmacological assay; Supplementary Table 1), with eight compounds in common with CCLE and GDSC (Supplementary Fig. 15). To check the predictive value of our biomarkers in breast cancer, we fitted the linear models in Eqs. (1–3) using only breast cancer cell lines in our training sets (61 and 54 breast cancer cell lines in CCLE and GDSC, respectively). A biomarker is selected if its predictive value in breast cancer cell lines is > 0.55. To validate the selected biomarkers in GRAY, we computed the significance of the linear association between the biomarker expression and drug response (unadjusted p-value < 0.05) with the same direction of association (sign of the coefficient $\beta$) as the training sets. To select the validated biomarkers whose isoform expression is significantly more predictive than the corresponding overall gene expression, we estimated the concordance index distribution of the isoform-based and gene-based models using the bootstrap procedure and compared these distributions using a two-sided Wilcoxon-signed rank test (Supplementary Fig. 13).

**Final validation of isoform-based biomarkers**. To test whether the predictive value of the isoform-based biomarkers validated in GRAY was robust to the use of a different pharmacological assay, we leveraged a collection of 84 breast cancer cell lines recently used to investigate gene essentiality in breast cancer molecular subtypes[21]. We selected 14 cell lines in this collection that were readily available and showed extreme expressions of the biomarkers of interest (Supplementary Table 1). Selected cell lines were cultured and screened for their response to three targeted agents: lapatinib, AZD6244 and erlotinib, and one chemotherapy, paclitaxel. We used the sulforhodamine B colorimetric (SRB) proliferation assay[60] in 96well plates to determine the drug dose–response curves. We subtracted the average phosphate-buffer saline (PBS) wells value from all wells and computed the standard deviation and coefficient for each triplicate. Data points with coefficient or standard deviation > 0.2 were discarded. All the individual treated well values were normalized to the control well values. We used the PharmacoGx[50] package to fit the curves using a logarithmic logistic regression method to estimate the AUC sensitivity values. Raw and processed pharmacological data are available through our PharmacoGx platform under the UHNBC PharmacoSet.

**Code availability**. Our code and documentation are open-source and publicly available through the RNAseqDrug GitHub repository (github.com/bhklab/RNASeqDrug). A detailed tutorial describing how to run our pipeline and reproduce our analysis results is available in the GitHub repository. Our study complies with the guidelines outlined in refs. [61, 62].

**Data availability**. The pharmacogenomics data used in this study are publicly available through our PharmacoGx platform[50]. CCLE and GDSC data are available from https://portals.broadinstitute.org/ccle/ and http://www.cancerrxgene.org/, respectively. The gCSI dataset is available from the European Genome-phenome Archive (EGAS00001000610). The GRAY dataset is available from EGA (EGAS00000000059 and EGAS00001000585) and ArrayExpress (E-TABM-157 and E-MTAB-181). The RNA-seq data for the UHN dataset are available on the

NCBI Gene Expression Omnibus (GSE73526), whereas the drug sensitivity data are available in PharmacoGx (UHNBreast).

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

## Acknowledgements

Z.S. was supported by the Terry Fox Research Institute and the Cancer Research Society. P.S. was supported by the Canadian Cancer Society Research Institute. B.H.-K. was supported by the Gattuso Slaight Personalized Cancer Medicine Fund at Princess Margaret Cancer Centre, the Canadian Institute of Health Research, Natural Sciences and Engineering Research Council, and the Stand Up To Cancer Canada. We like to thank the investigators of the Genomics of Drug Sensitivity in Cancer (GDSC), the Cancer Cell Line Encyclopedia (CCLE), Genentech (gCSI), Drs Joe W. Gray, and Benjamin G. Neel who have made their invaluable data available to the scientific community. The research was supported by a Stand Up To Cancer Canada – Canadian Cancer Society Breast Cancer Dream Team Research Funding, with supplemental support of the Ontario Institute for Cancer Research through funding provided by the Government of Ontario (Funding Award Number: SU2C-AACR-DT-18-15). Stand Up To Cancer Canada is a program of the Entertainment Industry Foundation Canada. Research funding is administered by the American Association for Cancer Research International - Canada, the Scientific Partner of SU2C Canada.

## Author contributions

Z.S. designed the study, collected the data, performed the data analysis and results interpretation, and wrote the manuscript. P.S. contributed to the data collection and analysis, results interpretation, and manuscript writing. K.L.T., J.S. and D.C. generated the UHNBreast pharmacological profiles and contributed to the results interpretation. N.E.-H. contributed to the data collection, results interpretation, and manuscript writing. R.Q. participated to the data collection and analysis. M.L. and T.W.M. contributed to the study design and results interpretation. D.C. contributed to the study design and manuscript writing. B.H.-K. designed and supervised the study.

## Additional information

**Competing interests:** The authors declare no competing financial interests.

**Change history:** A correction to this article has been published and is linked from the HTML version of this paper.

