## [Peer Review file · Nature Communications]

Reviewers' comments:

Reviewer #1 (Remarks to the Author):

The authors tried to identify biomarkers that are predictive for drug response in cancer cell lines based on gene expression data. Traditionally, the expression levels of genes were quantified at the gene level, i.e., one expression value was assigned for one gene. Because alternative splicing is widespread in human tissues, gene-level quantification of expressions may neglect important details at the transcript level. The authors tried to quantify gene expressions at the transcript level rather than the gene level in this study to identify biomarkers that are predictive of drug response in cancer cell lines. Although the idea is important and somewhat interesting, the results are expected and not surprising.

Major concerns:

1, In this study, the authors used AAC (Area Above the Curve) to summarize the data presented by drug dose-response curves. This is not convincing despite the listed justification to use AAC and references. Building models directly based on the drug dose-response curves rather than the summary statistics is more acceptable. Or at least, a series of summary statistics should be evaluated and compared to illustrate the impacts of summarization on modeling.

2, The authors used Tophat2 and Cufflinks to quantify the expression levels of human transcripts. Tophat2 and Cufflinks were once the leading tools to quantify transcript expression levels based on RNA-seq data. But with the rapid development of NGS analytical tools, new tools such as Hisat2 and StringTie that are much more powerful than Tophat2 and Cufflinks have been developed. The authors should update their quantification method for transcript expression levels to obtain more accurate results.

3, When expression levels were quantified by FPKM, the authors first applied a log₂ transformation and then did the modeling steps. Why did the authors apply this transformation at first? Is it for better normality of the expression distribution? If so, log₂(FPKM+1) may be not good enough. log₂(FPKM+10⁻ⁿ) with n more than 4 or 5 would yield better normality. The authors should also justify the necessity of the log₂ transformation. Is it needed by the subsequent analysis? Because FPKM has explicit biological meanings, it is better to demonstrate the superiority of log₂(FPKM+epsilon) over FPKM.

4, The authors fitted linear models to the drug response data. However, the R² values evaluating how well the models fit the data were not presented. With only p values, the readers were not informed sufficiently to evaluate the behaviors of the models.

5, Are there interactions between genes/transcripts and tissue types?

6, Based on the authors' experimental design, it can be expected that some genes are significantly predictive of drug response at both the gene and transcript levels, some genes are significantly predictive only at the transcript level, and some genes are significantly predictive only at the gene level. The results demonstrated by the authors were consistent with the expectations. However, the authors did not present the detailed information of these three groups of genes, especially of their splicing. Without the information, the readers have no idea of the factors that discriminate these three groups of genes.

Minor concerns:

1, How did the authors do multiple testing corrections? Two methods (Bonferroni and FDR) were listed in the manuscript. But which one was applied in practice?

2, Many figures were not well annotated. For example, in Figure 2A, the label for y-axis is "number of associated genes", but the bars are for both genes and isoforms. In Figure 4I and 4K, the median of tumor is less than that of healthy but was annotated as "Tumor>Healthy".

Reviewer #2 (Remarks to the Author):

The concept that gene isoforms could be used as drug response markers is an interesting and quite novel idea. The authors are also to be complimented for the use of two independent datasets for validation, something that is extremely important for such predictive modelling applications, as well as for publishing all their analysis codes to enable others to reproduce the results.

However, the current results, even if promising, lack coverage both in terms of drug compounds and cancer types considered, and the reported validation rates remains unfortunately quite modest even in this limited set. Finally, the clinical relevance of the presented results remains totally unknown in the absence of any clinical treatment response validations.

Major comments:

1) While the finding that gene isoforms capture more explanatory power than the gene expression patterns is indeed interesting, it is not too surprising, given that transcript-level data encodes much more biological information than the corresponding gene-level data. Therefore, it would be highly interesting to evaluate the true predictive power of gene isoforms (see comment 4 below) against other genomic and molecular information, such as point mutations, copy number alterations, DNA methylation changes or protein abundance differences, which have traditionally been used as predictive markers for drug response in cell line panels.

2) The evaluation is quite limited, starting with 15 drugs in the first discovery phase, then reducing to 8 drugs in the pre-validation phase, and eventually having only 4 drugs and 4 gene isoforms in the final validation phase. Moreover, the success rates in the different validations are quite modest, giving the impression of general lack of validation of these biomarkers. The two validation datasets are limited to breast cancer cell lines only, leaving the wider cancer relevance unknown. The authors are recommended to look for additional validation datasets, such as NCI-60 or Broad CTRP v2 resources, to make the results more systematic.

3) This study does not really evaluate the predictive power of the identified biomarkers and models, rather it reports whether the same markers (genes or their isoforms) show statistically significant correlation in the test datasets. This makes the gene vs. isoform comparisons and the validation results rather abstract. For instance, it remains unclear what is the biological or clinical relevance of >3000 biomarkers found for a particular drug, how many of the thousands of isoforms can even be expected to be validated in independent datasets, and how the biomarkers are linked to the targets, pathways or other mode-of-action mechanisms of the drugs?

4) The real predictive accuracy of the linear regression models remains unclear in the absence of any correlation plots that would quantify how accurately the model estimated using the training cell lines can actually predict drug response in the test cell lines (e.g., observed drug response vs. predicted response based on the test cell line RNA-seq data). Further, as shown in previous works (ref. 6), linear models and statistical p-value-based marker selection is unlikely to lead to optimal predictive models, instead non-linear models implementing regularized (penalized) feature selection are the state-of-the-art in the field.

5) The statistical methodology used seems overly-complicated and poorly-motivated. For instance, statistical significance levels are calculated using various procedures (uncorrected, Bonferroni- and FDR-corrected p-values) in the different validation phases, without providing rationale for these different options. Similarly, independent training and test set division has been done in the first validation phase, but implemented using bootstrap sampling, which introduces dependencies in the datasets. Standard, or ideally nested cross-validation (CV), is recommended, which keeps the two datasets independent across the CV rounds.

6) The results regarding the 4 selected isoforms in TCGA and GTEx datasets does not really

support their clinical relevance, since the expression distributions are completely overlapping between the tumor and healthy tissues. Even if there was a clear distinction between the two distributions, it remains unclear whether this also contributes to any differences in the clinical treatment responses. The authors should make an effort to find patient datasets where the clinical relevance of the isoform markers for treatment responses can be validated (e.g., TCGA or clinical studies that include actual treatment outcome data).

Minor comments:

1) R^2 is not really a metric of predictive power, rather it quantifies how well the linear regression model can explain the variation in a given dataset. Please change the manuscript text and title accordingly, or ideally, actually evaluate the predictive accuracy (see comment 4 above).

2) The linear models (Eqs. 2-3 in Supplement) effectively ignore any interactions between genes and isoforms, as well as interactions between tissue types and genes/isoforms. The authors should explain whether or not considering such interactions is relevant for drug response modelling.

3) The drug response and RNA-seq datasets originate from various sources. The authors should describe how they guaranteed that the cell lines are the same between the CCLE/GDSC and GRAY and UHN, and that there are no dependencies between the two discovery and two (independent) validation datasets.

4) Please state how many breast cancer cell lines were in the training datasets, and whether corrected p-values were used in the biomarker validation (Pre-validation Methods section).

5) This reviewer believes that the present methodology (linear models and p-value feature selection) leads to the massive number of identified markers (and effectively model over-fitting to small number of breast cancer cell lines), which at least partly explain the rather low validation success.

6) Figure 3 should show all the 8 drugs and give statistical significance between isoform vs. gene markers comparison.

Reviewer #1

The authors tried to identify biomarkers that are predictive for drug response in cancer cell lines based on gene expression data. Traditionally, the expression levels of genes were quantified at the gene level, i.e., one expression value was assigned for one gene. Because alternative splicing is widespread in human tissues, gene-level quantification of expressions may neglect important details at the transcript level. The authors tried to quantify gene expressions at the transcript level rather than the gene level in this study to identify biomarkers that are predictive of drug response in cancer cell lines. Although the idea is important and somewhat interesting, the results are expected and not surprising.

We thank the reviewer for her/his constructive comments. While it is expected that quantifying expression of specific isoforms can provide more information about the transcriptomic state of the cell line than summarizing this information at the overall gene level, we believe it is important to demonstrate that isoform expression quantification can improve the biomarker discovery process, a key step in fulfilling the promise of precision cancer medicine. To the best of our knowledge, this is the first study to investigate, at the genome-wide level, alternatively spliced isoforms as biomarkers across a number of datasets, drugs and cell lines, and we believe that, in our revised manuscript, we have firmly established the utility of isoform-level analysis for the future discovery of predictive biomarkers.

Major concerns:

1, In this study, the authors used AAC (Area Above the Curve) to summarize the data presented by drug dose-response curves. This is not convincing despite the listed justification to use AAC and references. Building models directly based on the drug dose-response curves rather than the summary statistics is more acceptable. Or at least, a series of summary statistics should be evaluated and compared to illustrate the impacts of summarization on modeling.

We appreciate the reviewer's concern regarding the use of a single measure to summarize all the information regarding the drug dose-response curve. We have previously observed that the point measurements of dose-response in the CCLE and GDSC studies are subject to substantial levels of noise (Safikhani et al. 2016), making each measure on its own unreliable for prediction. Fortunately, dose-response curves are known to follow a Hill slope model, fully

determined by a set of 3 parameters. The 8 (CCLE) or 9 (GDSC) concentrations tested in the studies reduce the variance in the estimates of the fitted curves compared to the variance of each single point, allowing for more precise quantification of the sensitivity of a cell line to drug treatments.

The Hill curves are often summarized by looking at one of three sensitivity measures, the IC_{50} , E_{inf} and AUC (1-AAC). The AUC has been shown to be the most consistent measure of drug response between independent studies (Fallahi-Sichani et al. 2013; Safikhani et al. 2016; Mpindi et al. 2016). Specifically, we have previously shown that inconsistencies in drug response profiling, especially when measured using the IC_{50} , leads to inconsistent selection of biomarkers between CCLE and GDSC. Maximizing the consistency between drug sensitivity measurements in these studies was paramount for our study given that the training of predictive biomarkers was done combining information from these two datasets. Therefore, we chose the AAC measure for quantifying response to drug treatment. Nevertheless, it is conceivable that other parameters can carry complementary information to the AAC, increasing the robustness of biomarkers selected between studies. We followed the reviewer's suggestion and investigated whether training predictive models using a combination of target summary measures could increase the consistency of biomarker selection between studies.

We computed the AAC, IC_{50} and E_{inf} measures for all experiments with the common drugs between CCLE and GDSC. We then used our previously published analysis pipeline (Smirnov et al. 2016; Safikhani et al. 2016) to examine consistency of significant gene expression - drug response (referred to as gene-drug) associations found in CCLE and GDSC, using a significance cutoff of $p < 0.05$. For both the CCLE and GDSC sensitivity data, we used the CCLE RNAseq gene expression profiles, so any disagreement in the gene-drug associations would be due to the differences in sensitivity data. We examined Matthew's Correlation Coefficient (MCC) between the selection of significant biomarkers to assess the consistency between the lists of significantly associated biomarkers. As expected from our previous work (Haibe-Kains et al. 2013; Safikhani et al. 2016), we found that the AAC metric was most consistent.

Significant Biomarkers ($p < 0.05$) MCC between GDSC and CCLE

We then computed MCC for multiple output models -- MANOVA as in (Garnett et al. 2012) -- with AAC, IC_{50} and E_{inf} measures as output variables to assess significance of the associations. We observed that MANOVA, despite being more complex and computational intensive than a simpler regression model, did not yield a significantly improved biomarker consistency over AAC alone.

Significant Biomarkers ($p < 0.05$) MCC between GDSC and CCLE

Given that using a multiple-output model brings increased complexity and requires greater statistical power (as the model doubles in complexity), we decided to use a single output

measure for the current study, leaving the exploration of using multiple output measures in biomarker discovery for future studies.

2, The authors used Tophat2 and Cufflinks to quantify the expression levels of human transcripts. Tophat2 and Cufflinks were once the leading tools to quantify transcript expression levels based on RNA-seq data. But with the rapid development of NGS analytical tools, new tools such as Hisat2 and StringTie that are much more powerful than Tophat2 and Cufflinks have been developed. The authors should update their quantification method for transcript expression levels to obtain more accurate results.

Next-generation sequencing analysis tools and pipelines are an active and quickly advancing area of research. Since the conception of our study there have been many methods published improving over Tophat2 and Cufflinks, of which, as the reviewer points out, Hisat2 and StringTie are currently leading examples. While preparing our revised results, we took the opportunity to update the pipeline used to quantify gene and isoform expression to Hisat2 and StringTie, and all results presented in our manuscript have been regenerated using values obtained from these tools. We also have updated the reference annotation file from Gencode version 12 to Gencode version 25 and GRCH38 reference genome. We looked into the consistency of the abundances of genes and isoforms in our old Tophat/Cufflinks based pipeline and the new estimations. We constrained the list of features to those which has been predicted as significantly associated with a drug response in our pipeline. As expected the genes are very well correlated between these pipelines while the expression of isoforms in the new pipeline has a lower correlation with the old estimations. This could be a result of the better annotation of transcripts in new reference annotation version as well as the more precise quantification of isoform expression by StringTie.

3, When expression levels were quantified by FPKM, the authors first applied a log2 transformation and then did the modeling steps. Why did the authors apply this transformation at first? Is it for better normality of the expression distribution? If so, $\log_2(\text{FPKM}+1)$ may be not good enough. $\log_2(\text{FPKM}+10^{-n})$ with n more than 4 or 5 would yield better normality. The authors should also justify the necessity of the log2 transformation. Is it needed by the subsequent analysis? Because FPKM has explicit biological meanings, it is better to demonstrate the superiority of $\log_2(\text{FPKM}+\epsilon)$ over FPKM.

As the reviewer points out, the $\log_2(\text{FPKM}+1)$ transformation is often applied in literature when investigating differential expression to help normalize the distribution of the expression values. Using the Hisat2+StringTie pipeline, we follow the guidelines published by Perteza et al. in applying the $\log_2(\text{FPKM}+1)$ transformation (Perteza et al. 2015; Perteza et al. 2016) before looking at the differential expression of isoforms and genes across different cell lines. To verify the utility of this transformation, we calculated the Shapiro Wilk statistic measuring normality of the distribution for each gene in the CCLE RNAseq data before and after the $\log_2(\text{FPKM}+1)$ transformation. The improvement in normality was found to be statistically significant (Wilcoxon signed rank $p < 1e-16$).

4, The authors fitted linear models to the drug response data. However, the R2 values evaluating how well the models fit the data were not presented. With only p values, the readers were not informed sufficiently to evaluate the behaviors of the models.

As both reviewers suggested that the predictive value (effect size) of the biomarkers of interest should be reported, we have revised our manuscript and supplementary information to include concordance index values for all biomarker models. The concordance index is a nonparametric predictive value estimate widely used in biomedical research (Harrell et al. 1996); it estimates the probability that, for a random pair of cell lines, a biomarker can rank them the same way than based on the sensitivity values measured experimentally. The nonparametric and interpretable nature of the concordance index were determinant in its selection for the recent drug sensitivity prediction DREAM challenge (Costello et al. 2014).

In order to estimate the range of concordance indices that is relevant for our de novo biomarker discovery, we investigated a set of known biomarkers we previously identified in the CCLE and GDSC datasets (Safikhani et al. 2016). As can be seen in Supplementary Table 4, the concordance index for approved biomarkers ranges from 0.55 to 0.68 at the isoform level. We therefore decided to select in our study all the biomarkers with concordance index ≥ 0.55 .

5, Are there interactions between genes/transcripts and tissue types?

In the context of biomarker discovery, a significant interaction between gene expression and tissue type would suggest a biomarker predictive of drug response within a specific tissue. Such biomarkers are indisputably of great interest, and many of the clinically relevant biomarkers for

drug response, such as ERBB2 expression for lapatinib in breast cancer, are employed within a specific cancer type. However, the goal of our study was to do an unbiased biomarker discovery across the full CCLE and GDSC datasets, and test these biomarkers using independent datasets. Unfortunately, while the CCLE and GDSC cell line panels are large when taken in aggregate, once they are stratified by tissue type we found that for all of the gene/transcript models in our training set, the sample size was insufficient to estimate all the expression-tissue interaction terms. On average, approximately a quarter of the tissue types in GDSC and CCLE did not have enough samples to estimate the interaction term between expression and tissue identity. Such a variability across genes clearly presents a problem, as it would lead to comparing models of different complexity between different genes and transcripts. Furthermore, such interactions would have to be verified in an independent dataset containing drug response across tissue types. While fortuitously the genentech Cell Screening Initiative (gCSI) was recently published, releasing independently generated molecular and drug sensitivity data, the smaller size of this dataset only exacerbates the difficulties with an insufficient number of samples per tissue. Given that the current available datasets do not contain sufficient sample sizes to adequately address the question of interactions between the terms of our model, we decided to limit our analysis to first order modeling of the dependence of drug response on mRNA expression.

6, Based on the authors' experimental design, it can be expected that some genes are significantly predictive of drug response at both the gene and transcript levels, some genes are significantly predictive only at the transcript level, and some genes are significantly predictive only at the gene level. The results demonstrated by the authors were consistent with the expectations. However, the authors did not present the detailed information of these three groups of genes, especially of their splicing. Without the information, the readers have no idea of the factors that discriminate these three groups of genes.

We investigated the discriminatory features between the “gene-specific”, “isoform-specific” and “common” features by first looking into the number of alternatively spliced isoforms for their corresponding gene (Supplementary Figure 11). As expected, the largest proportion of isoform-specific biomarkers are the product of the genes with multiple transcripts. We also categorized the predictive features by their biological types: protein coding, antisense, processed transcript, linc RNA, pseudogenes (Supplementary Figure 12). While the largest proportion of protein coding biomarkers are isoformic (isoform-specific and common), the processed transcript biomarkers is dominated by isoform specific ones. The pseudogene type contains the lowest ratio of isoform-specific biomarkers. We have updated our manuscript with these observations.

Minor concerns:

1, How did the authors do multiple testing corrections? Two methods (Bonferroni and FDR) were listed in the manuscript. But which one was applied in practice?

We first used bonferroni correction at the isoform level for each gene to ensure that the larger number of isoforms compared to genes is not conferring an unfair advantage to the latter. We therefore chose the most stringent family-wise error rate estimation approach to compute the p-value of the best isoform for each gene. We then used the well-established false discovery rate (FDR) approach to correct the “gene-level” p-values for multiple testing at the genome-wide level. Our multiple testing correction approach is now clearly illustrated in Supplementary Figure 2.

2, Many figures were not well annotated. For example, in Figure 2A, the label for y-axis is "number of associated genes", but the bars are for both genes and isoforms. In Figure 4I and 4K, the median of tumor is less than that of healthy but was annotated as "Tumor>Healthy".

We carefully reviewed all the figures and corrected the captions and axis labels. We have now clarified that the label of y-axis in Figure 2A refers to the fact that gene and isoform models are compared at gene level to emphasize that our results is not due to the larger number of isoforms.

Reviewer #2

The concept that gene isoforms could be used as drug response markers is an interesting and quite novel idea. The authors are also to be complimented for the use of two independent datasets for validation, something that is extremely important for such predictive modelling applications, as well as for publishing all their analysis codes to enable others to reproduce the results.

However, the current results, even if promising, lack coverage both in terms of drug compounds and cancer types considered, and the reported validation rates remains unfortunately quite modest even in this limited set. Finally, the clinical relevance of the presented results remains totally unknown in the absence of any clinical treatment response validations.

We thank the reviewer for her/his positive feedback regarding our manuscript. We agree with the reviewer that a pan-cancer validation of more drugs would strengthen our study. We have therefore included the very recent Genentech Cell Line Screening Initiative (gCSI) dataset (Haverty et al. 2016), which we integrated in our PharmacoGx platform. This dataset contains 16 drugs, which 5 are overlapping with our training set (Supplementary Figure 7 in updated manuscript). Importantly, this dataset is unique as it is the only dataset that contains both RNA-seq data and drug sensitivity across multiple tissue types (Supplementary Table 1), beside CCLE. Thanks to the large sample size of gCSI, we observed high validation rate for multiple drugs (61% and 54% for erlotinib and lapatinib, respectively; see Figure 3). These new results clearly indicate that reasonable validation rate can be achieved when a large validation set is available.

We agree with the reviewer that demonstrating clinical relevance of biomarkers discovered in vitro is an important research endeavor. Such a clinically-oriented validation could be performed first in vivo -- in patient-derived xenografts (PDX) for instance -- and then in clinical samples, using materials from clinical trials in a neoadjuvant setting. Although outside the scope of our study (limited to in vitro models), we have inquired the Novartis PDX Encyclopedia team to get access to their pharmacogenomic data but they declined to release the raw RNA-seq files, which are required to run our pipeline (Joshua Korn; personal communication). We have also looked for clinical trials with Response Evaluation Criteria in Solid Tumors (RECIST) information for drugs in our study and, to the best of our knowledge, no such datasets are available. In this context, although we agree with the reviewer that further in vivo preclinical and clinical validations are necessary to assess the relevance of our new in vitro biomarkers, we believe such experiments and analyses are outside the scope of the present study.

Major comments:

1) While the finding that gene isoforms capture more explanatory power than the gene expression patterns is indeed interesting, it is not too surprising, given that transcript-level data encodes much more biological information than the corresponding gene-level data. Therefore, it would be highly interesting to evaluate the true predictive power of gene isoforms (see comment 4 below) against other genomic and molecular information, such as point mutations, copy number alterations, DNA methylation changes or protein abundance differences, which have traditionally been used as predictive markers for drug response in cell line panels.

To address the reviewer's comment, we compared the predictive value of the gene/isoform expression against the additional molecular profiles in CCLE, i.e, the copy number alteration and mutation data. Concurring with the Drug Sensitivity Prediction DREAM challenge (Costello et al. 2014), our results support the superiority of expression-based features for drug sensitivity prediction in vitro (Supplementary Figure 5). Overall, there were significantly more isoform-based biomarkers than gene expression and copy number alterations (one-sided Wilcoxon rank sum test p -value < 0.001). However, even though there were more isoform-based biomarkers than mutation-based biomarkers for the majority of the drugs, mutations were more predictive for nilotinib and crizotinib (Supplementary Figure 5). Altogether, these new results support the isoform expression as a promising new class of biomarkers for the majority of the drugs.

2) The evaluation is quite limited, starting with 15 drugs in the first discovery phase, then reducing to 8 drugs in the pre-validation phase, and eventually having only 4 drugs and 4 gene isoforms in the final validation phase. Moreover, the success rates in the different validations are quite modest, giving the impression of general lack of validation of these biomarkers. The two validation datasets are limited to breast cancer cell lines only, leaving the wider cancer relevance unknown. The authors are recommended to look for additional validation datasets, such as NCI-60 or Broad CTRP v2 resources, to make the results more systematic.

We agree that a validation across different cancer types is interesting in the context of biomarker discovery, especially since the model used to discover biomarkers in our study was trained across multiple cancer types. The venerable NCI60 dataset includes a very large set of drugs but is limited to 60 cell lines, making inadequate for validation of pan-cancer biomarkers in vitro. The CTRPv2 dataset is spanning 860 cell lines across many cancer types although lacking RNA-seq profiles, which is required for our validation pipeline. Fortunately, a recently released dataset, the Genentech Cell Line Screening Initiative (gCSI) (Haverty et al. 2016) is perfectly poised to address this issue, as it contains independently generated RNA-seq data as well as drug sensitivity measurements for over 400 cell lines (Supplementary Figure 7). This allowed us to update our manuscript with a pan-cancer validation of our predictive biomarkers (Figure 3). We updated the manuscript to report these new pan-cancer results.

3) This study does not really evaluate the predictive power of the identified biomarkers and models, rather it reports whether the same markers (genes or their isoforms) show statistically significant correlation in the test datasets. This makes the gene vs. isoform comparisons and the validation results rather abstract. For instance, it remains unclear what is the biological or clinical relevance of >3000 biomarkers found for a particular drug, how many of the thousands of isoforms can even be expected to be validated in independent datasets, and how the biomarkers are linked to the targets, pathways or other mode-of-action mechanisms of the drugs?

We have updated our methodology to include a filtering of the biomarkers by their predictive value (concordance index), as described in more detail in Supplementary Figure 2. We have also added the new gCSI dataset, in which we repeated the bootstrapping procedure to select biomarkers showing significant improvements in predictive value over the null model (containing only a tissue term). The results in gCSI allowed us to estimate the validation rates of isoform-specific biomarkers using independent pan-cancer datasets, which can be as high as 61% and 54% for erlotinib and lapatinib (Figure 3 and Supplementary Table 2). Actually, we observed that isoform-specific biomarkers consistently yield a higher validation rate than gene-specific biomarkers, supporting our rationale that isoforms represent a promising class of biomarkers.

We agree with the reviewer that the thousands of biomarkers found for the MEK inhibitor PD-0325901 is unexpected. We observed that (i) this drug yielded a large range of AAC values across the cell lines in both CCLE and GDSC, and that (ii) the drug sensitivity consistency between training sets was the highest among drugs undergoing validation (Supplementary Figure 8 and Figure 3 of updated manuscript), contributing to the large number of significant biomarkers. Interestingly, 41% of isoform-based biomarkers could be validated in gCSI, against 15% for gene-specific biomarkers. Our results suggest that, when the pharmacological profile are moderately consistent with a wide range of drug sensitivities, one can validate many biomarkers for PD-0325901. Although it is tempting to extrapolate this anecdotal observation to other (targeted) drugs, our drug panel is not large enough to support such a claim. While we

agree that carrying out mechanistic studies on the validated biomarkers would be of interest to uncover the functional links with the drug targets, we believe that these experiments are outside the scope of our study.

4) The real predictive accuracy of the linear regression models remains unclear in the absence of any correlation plots that would quantify how accurately the model estimated using the training cell lines can actually predict drug response in the test cell lines (e.g., observed drug response vs. predicted response based on the test cell line RNA-seq data). Further, as shown in previous works (ref. 6), linear models and statistical p-value-based marker selection is unlikely to lead to optimal predictive models, instead non-linear models implementing regularized (penalized) feature selection are the state-of-the-art in the field.

As suggested by both reviewers, we have updated the manuscript to report the predictive value (concordance index) for the bootstrap procedure for each biomarker (see comment #4 of Reviewer #1). These values are computed on the data held out from training during each bootstrap iteration (~1/3 of available cell lines), giving an estimate of the expected predictive value of a given biomarker candidate. For the pan-cancer validation using gCSI (see response to comment #2 above), we repeated this procedure and once again reported the median concordance index. We considered biomarkers validated if they achieve both a significant increase in predictive accuracy over the null model (only tissue type), and have a concordance index >0.55 in the validation set (see comment #4 of Reviewer #1). We have also updated our supplementary data with scatterplots of the predicted vs actual AAC values after controlling for tissue type, using our top selected biomarkers (Supplementary Figure 10 and Supplementary Figure 14).

While we agree that non-linear models implementing penalized feature selection may in some contexts offer an advantage in predictive accuracy over simpler linear modeling techniques, it is not clear that these models would be reproducible between datasets. The stability of penalized feature selection between datasets is specifically in question. Work by Haury et al. (Haury et al. 2011) examined the effect of different feature selection methods on the ability to use microarray expression to predict breast cancer prognosis. While filtering by t-test associations, lasso and elastic net feature selection were the only methods found to outperform random feature selection in terms of model accuracy, when assessing stability between selected features between datasets, of these three methods only t-test based filtering was able to outperform randomly selected feature sets. This study underlines the well known difficulty of robust feature selection in the genomic domain (Ein-Dor et al. 2006), an impediment to a more systematic application of multivariate modeling in the context of pharmacogenomics.

5) The statistical methodology used seems overly-complicated and poorly-motivated. For instance, statistical significance levels are calculated using various procedures (uncorrected, Bonferroni- and FDR-corrected p-values) in the different validation phases, without providing rationale for these different options. Similarly, independent training and test set division has been done in the first validation phase, but implemented using bootstrap sampling, which

introduces dependencies in the datasets. Standard, or ideally nested cross-validation (CV), is recommended, which keeps the two datasets independent across the CV rounds.

We have updated the manuscript to clearly describe the training and validation sets (Figure 1) as well as the training procedure to identify our initial set of isoform-based biomarker candidates (Supplementary Figure 2). We first used bonferroni correction at the isoform level for each gene to ensure that the larger number of isoforms compared to genes is not conferring an unfair advantage to the latter. We therefore chose the most stringent family-wise error rate estimation approach to compute the p-value of the best isoform for each gene. We then used the well-established false discovery rate (FDR) approach to correct the “gene-level” p-values for multiple testing at the genome-wide level. We agree with the reviewer that dependencies exist across the bootstrap runs, and that is why we refer to our approach as a “pre-validation” framework as published by Hofling and Tibshirani (2008), and we relied on multiple independent validation sets to test the robustness of our biomarker candidates.

6) The results regarding the 4 selected isoforms in TCGA and GTEx datasets does not really support their clinical relevance, since the expression distributions are completely overlapping between the tumor and healthy tissues. Even if there was a clear distinction between the two distributions, it remains unclear whether this also contributes to any differences in the clinical treatment responses. The authors should make an effort to find patient datasets where the clinical relevance of the isoform markers for treatment responses can be validated (e.g., TCGA or clinical studies that include actual treatment outcome data).

We agree with the reviewer that the comparison of the biomarker distributions in tumor and normal tissues does not guarantee clinical relevance. Given that we used a new pipeline (Hista2 + StringTie) suggested by Reviewer #1, we could not directly compare the TCGA and GTEx data as they were processed with different pipelines, potentially adding substantial bias. We therefore removed this comparison from our study for the sake of statistical rigor and clarity. We also agree that isoform-based biomarkers should be validated in vivo to demonstrate their clinical relevance. As mentioned previously, RNA-seq profiles of patients treated with one of the drugs used in our study, and with RECIST criterion in neoadjuvant settings, are not currently available. We then inquired the Novartis PDX Encyclopedia team to get access to their pharmacogenomic data but they declined to release the raw RNA-seq files, which are required to run our pipeline (Joshua Korn; personal communication). In this context, although we agree with the reviewer that further in vivo preclinical and clinical validations are necessary to assess the relevance of our new in vitro biomarkers, we believe that such experiments and analyses are outside the scope of the present study.

Minor comments:

1) R^2 is not really a metric of predictive power, rather it quantifies how well the linear regression model can explain the variation in a given dataset. Please change the manuscript

text and title accordingly, or ideally, actually evaluate the predictive accuracy (see comment 4 above).

As mentioned above in our reply to comment #4, we updated our manuscript to report the predictive value, as estimated by the concordance index, of known (clinically relevant) biomarkers as well as our novel isoform-based candidate biomarkers (Supplementary Tables 4 and 5, Supplementary Files 4, 6).

2) The linear models (Eqs. 2-3 in Supplement) effectively ignore any interactions between genes and isoforms, as well as interactions between tissue types and genes/isoforms. The authors should explain whether or not considering such interactions is relevant for drug response modelling.

In the context of biomarker discovery, a significant interaction between gene expression and tissue type would suggest a biomarker predictive of drug response within a specific tissue (see comment #5 of Reviewer #1). Such biomarkers are indisputably of great interest, and many of the clinically relevant biomarkers for drug response, such as ERBB2 expression for lapatinib in breast cancer, are employed within a specific cancer type. However, the goal of our study was to do an unbiased biomarker discovery across the full CCLE and GDSC datasets, and test these biomarkers using independent datasets. Unfortunately, while the CCLE and GDSC cell line panels are large when taken in aggregate, once they are stratified by tissue type we found that for all of the gene/transcript models in our training set, the sample size was insufficient to estimate all the expression-tissue interaction terms. On average, approximately a quarter of the tissue types in GDSC and CCLE did not have enough samples to estimate the interaction term between expression and tissue identity. Depending on the expression of specific transcripts, this sometimes dropped to none of the interaction terms having enough unique expression values to estimate the interaction. The variability across genes clearly presents a problem, as it would lead to comparing models of different complexity between different genes and transcripts. Furthermore, such interactions would have to be verified in an independent dataset containing drug response across tissue types. While fortuitously the genentech Cell Screening Initiative (gCSI) was recently published, releasing independently generated molecular and drug sensitivity data, the smaller size of this dataset only exacerbates the difficulties with an insufficient number of samples per tissue. Given that the current available datasets do not contain sufficient sample sizes to adequately address the question of interactions between the terms of our model, we decided to limit our analysis to first order modeling of the dependence of drug response on mRNA expression.

3) The drug response and RNA-seq datasets originate from various sources. The authors should describe how they guaranteed that the cell lines are the same between the CCLE/GDSC and GRAY and UHN, and that there are no dependencies between the two discovery and two (independent) validation datasets.

In our previous study (Safikhani et al. 2016), we performed an in-depth comparison of the SNP fingerprints between GDSC and CCLE and confirmed that all the cell lines, except 4 (SWA403, COR-L51, MOG-G-CCM and NB4), have matched identity between the two datasets. We performed the same analysis for gCSI, GRAY and UHN datasets, and removed all the cell lines for which the SNP fingerprints did not match across datasets. We updated the Supplementary Information with these important results. We compared the checksum of all the RNA-seq files across datasets to ensure that there are no duplicated profiles in our study.

4) Please state how many breast cancer cell lines were in the training datasets, and whether corrected p-values were used in the biomarker validation (Pre-validation Methods section).

We apologize for this omission. We now clearly state the number of breast cancer cell lines in the training set (61 and 54 breast cancer cell lines in CCLE and GDSC, respectively; Supplementary Figure 6) and the fact that we relied on unadjusted p-values (<0.05) to estimate the validation rate in the independent datasets (pre-validation phase).

5) This reviewer believes that the present methodology (linear models and p-value feature selection) leads to the massive number of identified markers (and effectively model over-fitting to small number of breast cancer cell lines), which at least partly explain the rather low validation success.

Thanks to the constructive comments of the reviewers, we have incorporated a filter based on predictive value (concordance index > 0.55 , threshold based on the set of known biomarkers listed in Supplementary Table 4). This, in addition to the reprocessing of the RNA-seq data using the new Hista2 and StringTie pipeline, led to a global reduction of the number of candidate biomarkers for the vast majority of the drugs (Figure 2). The MEK inhibitor is, however, an exception, with over 1000 significant biomarker with concordance index > 0.55 . We noticed that (i) this drug yielded a large range of AAC values across the cell lines in both CCLE and GDSC, and that (ii) the drug sensitivity consistency was the highest between training sets among drugs undergoing validation (Supplementary Figure 10), contributing to the large number of significant biomarkers. Interestingly, 43% of isoform-based biomarkers could be validated in gCSI, against 18% for gene-specific biomarkers. Similarly, we could validate many of our pan-cancer candidate biomarkers in the new gCSI dataset, with 63% and 56% validation rates for erlotinib and lapatinib, although this was not the case for all drugs (only 9% for paclitaxel due to the lack of consistency of drug sensitivity data between CCLE, GDSC and gCSI; see Figure 3 in the updated manuscript). Our new results on the gCSI dataset therefore support larger proportion of validated biomarkers in a pan-cancer setting compared to breast cancer-specific biomarkers. While we agree that overfitting on the training set cannot be excluded, our findings indicate that a test set with larger sample size, such as gCSI, are required to yield a higher validation rate.

6) Figure 3 should show all the 8 drugs and give statistical significance between isoform vs. gene markers comparison.

We updated Figure 3 and Supplementary Figure 13 to highlight the biomarkers that are significantly better at the isoform level or gene level for the 8 drugs in the GRAY validation set. P-values are reported in Supplementary File 5.

Reviewers' comments:

Reviewer #1 (Remarks to the Author):

The authors have addressed all my methodological concerns. But the conclusions "The results of our meta-analysis of pharmacogenomic data suggest that isoforms represent a rich resource for biomarkers predictive of response to chemo- and targeted therapies. Our study also showed that the validation rate for this type of biomarkers is low (<50%) for most drugs, supporting the requirements for independent datasets to identify reproducible predictors of response to anticancer drugs." are somehow weak. The take n-home message is not so novel.

Reviewer #2 (Remarks to the Author):

The authors have made a good job in addressing the original comments and questions. However, some of the comments were addressed only partly and there remain some open questions:

- Calculating the concordance index values makes the evaluation of the predictive power easier, but why these results were placed into the supplement? Some of these new results seem also unexpected: why the null model c-indices are >0.50 (random model, Suppl. Table 4), and why the validation set c-indices are so much higher than the training set c-indices (Suppl. Table 5)?
- Comparison of the isoform-based markers with those from gene expression, mutations and amplifications was done based merely on the number of significantly associated markers (Suppl. Fig. 5), which cannot really compare their predictive power or value. Why not to show the distribution of concordance index for each dataset separately?
- The statistical testing part is now clearer, but like the authors admit, the 100 bootstrapped c-index values cannot be really treated as independent samples (page 7), making the p-values calculated based the Wilcoxon tests somewhat inflated. This should be clearly stated in the manuscript text, and ideally addressed in the analysis pipeline.
- This reviewer would like to see correlation plots to quantify how accurately the model estimated using the training cell lines can actually predict drug response in the test cell lines (e.g., observed drug response vs. predicted response based on the test cell line RNA-seq data). The residual plots in Suppl. Fig. 10 make such evaluation difficult.

Minor comments:

There were a number of typos in the added/modified text (yellow-highlights) and supplementary figure legends that should be corrected.

Reviewer #1

The authors have addressed all my methodological concerns. But the conclusions "The results of our meta-analysis of pharmacogenomic data suggest that isoforms represent a rich resource for biomarkers predictive of response to chemo- and targeted therapies. Our study also showed that the validation rate for this type of biomarkers is low (<50%) for most drugs, supporting the requirements for independent datasets to identify reproducible predictors of response to anticancer drugs." are somehow weak. The take-home message is not so novel.

To clearly state the novelty of our study, we updated the take home message with the following three points. (1) Although several published studies recently highlighted the difficulties in identifying robust biomarkers in vitro, we show for the first time how multiple large-scale pharmacogenomic datasets can be integrated for both biomarker discovery and validation. (2) We are also the first to investigate, at a genome-wide level, the expression of gene isoforms as a new class of biomarkers associated to sensitivity to targeted and chemotherapies. (3) we found that specific isoforms of IGF2BP2, NECTIN4, ITGB6 and KLHDC9 were significantly associated with AZD6244, lapatinib, erlotinib and paclitaxel, respectively, in multiple screening using different pharmacological assays, supporting their robustness in vitro. Our study provides a new analytical framework to identify promising isoform-based biomarkers, which will need to be further validated in vivo and in clinical settings.

Reviewer #2

The authors have made a good job in addressing the original comments and questions. However, some of the comments were addressed only partly and there remain some open questions:

- Calculating the concordance index values makes the evaluation of the predictive power easier, but why these results were placed into the supplement? Some of these new results seem also unexpected: why the null model c-indices are >0.50 (random model, Suppl. Table 4), and why the validation set c-indices are so much higher than the training set c-indices (Suppl. Table 5)?

The null model is assessing the predictive power of tissue types for drug responses. The activation of drug target pathways might be different across various tissue types resulting in some association between drug response and tissue type. Concurring with previous studies

(Garnett et al, Nature 2012; Haibe-Kains et al Nature 2013; Iorio et al Cell 2016, Yao et al JAMIA 2017), we observed that tissue type alone is significantly associated with sensitivity to many drugs, yielding concordance indices greater than 0.50 for the “null” model. We have updated the manuscript and the caption of Supplementary Table 4 to clearly state that the null model refers to the model with tissue type as sole predictor.

We thank the reviewer for pointing out that we should report the concordance indices for breast cancer in the training set for a fair comparison with the results from the validation sets. We updated the Supplementary Table 5 accordingly. The reason why the concordance indices are sometimes higher in the validation set lies in the fact that the validation sets are limited to breast cancer. In this situation, the molecular heterogeneity is reduced, leading to higher concordance indices for the majority of the biomarkers. We have updated the table caption to clearly describe these results.

- Comparison of the isoform-based markers with those from gene expression, mutations and amplifications was done based merely on the number of significantly associated markers (Suppl. Fig. 5), which cannot really compare their predictive power or value. Why not to show the distribution of concordance index for each dataset separately?

Following the reviewer’s suggestion, we added Supplementary Figure 9B to report the distribution of the concordance indices for the gene, isoform expression, mutations and copy number variations for each drug separately.

- The statistical testing part is now clearer, but like the authors admit, the 100 bootstrapped c-index values cannot be really treated as independent samples (page 7), making the p-values calculated based the Wilcoxon tests somewhat inflated. This should be clearly stated in the manuscript text, and ideally addressed in the analysis pipeline.

We agree with the reviewer and we have updated the manuscript to clearly state this limitation. We have also contacted Drs. Robert Tibshirani and Stefan Wager (Stanford University) on this issue. It appears that taking into account the dependency across bootstraps is a rather challenging statistical problem and a new method must be developed to handle it properly. Given that many other studies suffer from this limitation, we started a collaboration on this important matter and this will be the topic of a future publication.

- This reviewer would like to see correlation plots to quantify how accurately the model estimated using the training cell lines can actually predict drug response in the test cell lines (e.g., observed drug response vs. predicted response based on the test cell line RNA-seq data). The residual plots in Suppl. Fig. 10 make such evaluation difficult.

To address the reviewer’s comments, we have added Supplementary Figure 11 where we show observed drug response vs. predicted response for the tissue type with the strongest interaction

in CCLE, GDSC and gCSI for the top biomarker for each drug. For completeness, we also added the same plots for all the tissues in Supplementary File 5.

Minor comments:

There were a number of typos in the added/modified text (yellow-highlights) and supplementary figure legends that should be corrected.

We have carefully reviewed the manuscript and corrected all the typos.

REVIEWERS' COMMENTS:

Reviewer #2 (Remarks to the Author):

The new Suppl. Fig. 9 nicely shows why the number of significant predictors is not an optimal measure for 'predictive accuracy', as can be seen by comparing the panels A and B of Suppl. Fig. 9 (although a density plots of the c-indices would make the differences between the various marker types even clearer, compared to the current histogram plot). Anyways, based on the histograms, it seems that in addition to nilotinib and crizotinib (which were mentioned in the text, p. 10), also PLX-4720, lapatinib, AZD6244, TAE684 and PD-0325901 are in fact better predicted with mutations than with isoform-based markers. The authors should discuss this interesting observation more in the text, as it opens up the possibility to design even more accurate predictive models based on combining isoform and mutation-based expression panels. Mutations are currently being used as predictive markers for specific cancer treatments, so having additional predictors, even if this requires isoform-level assays, would be interesting from the translational point of view. Having a minimal panel of maximally predictive markers (with large effect sizes) is much more practical in terms of clinical assay designs, as compared to having large panels of less-predictive, yet significant predictors, which would require more expensive clinical assays. The authors should also consider adding Suppl. Fig. 9 as one of the main figures (with histograms replaced by non-filled density distribution plots), as this figure is much more informative about the predictive accuracies, compared to the 'number of predictors' plots in the present main figures.

Minor comments and typos:

- Suppl. Fig. 9, Panel A, color legend: Amlifications' -> Amplifications; Panel B, caption: significantly higher THAN for genes(?); please delete the second (B).
- Suppl. Fig. 11, consider making these plots with all the significant biomarkers, as the current scatter plots with the top-marker only does not look too convincing.
- Fig. 3, caption: please define AAC in the caption text to make it self-explanatory; consider using AAC also in Fig. 4 and 5 for consistency; panel D, the grey bars are not specified in the color legend.
- Fig. 4 and 5, please write AUC/AAC in the y-axis of dot-plots. The c-index values in the side bar plots of Fig. 4 seem too low (random classifier already gives c-index of 0.5).
- In many figures and tables, 'lapatinib' and 'paclitaxel' are written in non-capitalized first letter, whereas the other compounds are capitalized.
- Suppl. Table 1, caption: if -> of
- Suppl. Table 5, the validation set c-indices seem still too high, when compared to the breast cancer training set c-indices; please double-check your calculations.

Reviewer #2

The new Suppl. Fig. 9 nicely shows why the number of significant predictors is not an optimal measure for 'predictive accuracy', as can be seen by comparing the panels A and B of Suppl. Fig. 9 (although a density plots of the c-indices would make the differences between the various marker types even clearer, compared to the current histogram plot). Anyways, based on the histograms, it seems that in addition to nilotinib and crizotinib (which were mentioned in the text, p. 10), also PLX-4720, lapatinib, AZD6244, TAE684 and PD-0325901 are in fact better predicted with mutations than with isoform-based markers. The authors should discuss this interesting observation more in the text, as it opens up the possibility to design even more accurate predictive models based on combining isoform and mutation-based expression panels. Mutations are currently being used as predictive markers for specific cancer treatments, so having additional predictors, even if this requires isoform-level assays, would be interesting from the translational point of view. Having a minimal panel of maximally predictive markers (with large effect sizes) is much more practical in terms of clinical assay designs, as compared to having large panels of less-predictive, yet significant predictors, which would require more expensive clinical assays. The authors should also consider adding Suppl. Fig. 9 as one of the main figures (with histograms replaced by non-filled density distribution plots), as this figure is much more informative about the predictive accuracies, compared to the 'number of predictors' plots in the present main figures.

We have now included Suppl Fig 9 as Figure 3 in the main text. We agree with the reviewer on the interpretation of the results and we have updated the text to reflect on these points.

Minor comments and typos:

- Suppl. Fig. 9, Panel A, color legend: Amlifications' -> Amplifications; Panel B, caption: significantly higher THAN for genes(?); please delete the second (B).

Corrected.

- Suppl. Fig. 11, consider making these plots with all the significant biomarkers, as the current scatter plots with the top-marker only does not look too convincing.

Given the large number of significant biomarkers and tissue types, generating these figures for all significant biomarkers would result in a document with approximately 3000 pages, making it irrelevant for the readers. However, we made our code publicly available and fully documented, therefore allowing readers to easily generate these plots.

- Fig. 3, caption: please define AAC in the caption text to make it self-explanatory; consider using AAC also in Fig. 4 and 5 for consistency; panel D, the grey bars are not specified in the color legend.

Corrected.

- Fig. 4 and 5, please write AUC/AAC in the y-axis of dot-plots. The c-index values in the side bar plots of Fig. 4 seem too low (random classifier already gives c-index of 0.5).

Figures 4 and 5 are updated with AAC added as y-label of the plots. The reviewer is right about the c-index values, the plots are actually show the deviation of c-index values from 0.5 as the origin. This has been clarified.

- In many figures and tables, 'lapatinib' and 'paclitaxel' are written in non-capitalized first letter, whereas the other compounds are capitalized.

We made sure all the compound names are non-capitalized in updated version of the manuscript aside for those with abbreviation in their names.

- Suppl. Table 1, caption: if -> of

Corrected.

- Suppl. Table 5, the validation set c-indices seem still too high, when compared to the breast cancer training set c-indices; please double-check your calculations.

We double checked all our calculations and they are correct to the best of our knowledge. The fact that half the biomarkers yielded higher concordance index in breast cancer cell lines compared to pan-cancer estimates (training cindex) is due to the reduced molecular heterogeneity in specific tissue type (breast cancer).